# MITIGATING COPY BIAS IN IN-CONTEXT LEARNING THROUGH NEURON PRUNING

## ABSTRACT

Large language models (LLMs) have demonstrated impressive few-shot in-context learning (ICL) abilities. Still, we show that they are sometimes prone to a 'copying bias', where they copy answers from provided examples instead of learning the underlying patterns. In this work, we propose a novel and simple method to mitigate such copying bias. First, we create a synthetic task and use the Integrated Gradients method to identify neurons that prioritize copying over generalization. We demonstrate that pruning these neurons consistently improves performance across a diverse set of ICL tasks. We also show that our method is applicable across various LLM architectures, including Transformers and State-Space Models, without requiring modifications. In our analysis, we adopt a task-recognition perspective on ICL and examine task vectors (Hendel et al., 2023) induced by the model. We find that pruning enhances the quality of these vectors, suggesting that the pruned neurons previously hindered effective task recognition.

## 1 INTRODUCTION

In-Context Learning (ICL) (Brown et al., 2020) has recently emerged as a powerful and simple alternative to traditional training and fine-tuning. ICL involves presenting a Large Language Models (LLM) with a "context" consisting of several example pairs, each containing an input and its corresponding correct output, followed by a test example for prediction. For instance, consider the following prompt:

$$\text{Dolphin} \rightarrow 2, \quad \text{Beautiful} \rightarrow 5, \quad \text{Octopus} \rightarrow$$

In this case, the model must leverage the contextual information from the given examples to identify the underlying pattern of mapping words to vowel counts; based on this, it must then predict that the correct answer for "Octopus" is "3".

While ICL has shown considerable effectiveness, its use in few-shot scenarios faces significant challenges (Zhao et al., 2021; Razeghi et al., 2022). In these settings, the inherent scarcity of labeled examples becomes a critical bottleneck, as ICL often requires a substantial number of in-context examples to generalize effectively. Moreover, the performance of ICL is highly sensitive to various aspects of the prompt and the presentation of the examples. Factors such as the specific wording of the prompt (Wang et al., 2024), the order in which the examples are presented (Lu et al., 2021), and their relevance to the target example can significantly influence the outcome. Consequently, in domains where labeled data is limited, these challenges collectively hinder the reliable application of ICL, emphasizing the need for strategies that can mitigate sensitivity and make the most of the scarce examples available.

Recent research has primarily addressed these challenges by focusing on prompt formulation strategies, including techniques for selecting optimal templates and examples (Zhou et al., 2023b; Hao et al., 2022; Lu et al., 2022), as well as calibration methods (Han et al., 2023; Zhao et al., 2021). However, existing work has not yet explored how errors in in-context learning (ICL) relate to the internal processes of LLMs or how to correct them through targeted model modifications.

Our study takes a novel approach by investigating neural activation patterns related to a common challenge in ICL: copying errors. Referring back to the vowel-counting example, a copying error would occur if the model were to output "2" or "5" for Octopus, instead of the correct answer "3".

In these cases, the model appears to directly copy an answer from the provided examples rather than generating the correct, novel response based on the induced underlying pattern.

In this work, we hypothesize that there is a small subset of neurons in language models that prioritize copying behavior over task recognition. We posit that these mechanisms can be task-agnostic; that is, the same neurons are responsible for this reasoning shortcut across a range of tasks. We further hypothesize that deactivating these neurons will make the model less likely to follow shortcuts and encourage it to focus on recognizing underlying regularities.

To identify these neurons, we employ the vowel-counting task and apply the attribution method, Integrated Gradients (IG) (Sundararajan et al., 2017), to trace the copying errors to individual neurons. We then select the top contributing neurons as "copying neurons." The vowel-counting task is particularly interesting, as it appears challenging for a range of models and elicits the copying behavior in these models. We demonstrate that deactivating the neurons identified using this single task improves results across a diverse range of ICL problems, making our method practical – since the neurons do not need to be selected for each individual task – and confirming the existence of a general mechanism prioritizing the shortcut over reasoning.

In summary, our contributions are fourfold: (1) We identify the copying bias in ICL and demonstrate that LLMs, particularly smaller ones, suffer from a high percentage of these errors. (2) We introduce a method to identify specific neurons responsible for triggering the copying behavior (copying neurons). (3) We show that pruning these identified neurons leads to out-of-the-box improvement across a wide range of ICL tasks. (4) We utilize the task vectors framework introduced by (Hendel et al., 2023), quantifying a model's ability to recognize and adapt to tasks during ICL. Using this framework, we provide evidence that pruning the copying neurons enhances the quality of task vectors, indicating improved task representation. This finding explains the observed performance gains across various ICL tasks.

## 2 RELATED WORK

ICL, first introduced by (Brown et al., 2020), has attracted significant interest in recent years due to its ability to enable large language models to perform complex downstream tasks without explicit fine-tuning. By leveraging contextual information provided within the input prompts, these models can dynamically adapt their behavior and generate contextually relevant outputs across a wide range of tasks. While it is commonly associated with Transformer architectures, ICL has also been explored in other model architectures, such as State-Space Models and the RWKV model (Grazzi et al., 2024b; Park et al., 2024). Leading to a wide line of works that seek to improve the effectiveness of ICL mechanisms (Zhao et al., 2021; Wei et al., 2023; Chu et al., 2023; Zhou et al., 2023a; Wei et al., 2024; Li et al., 2024), as well as studies that aim to explain the underlying processes and dynamics of how models internalize and utilize context (Min et al., 2022; Liu et al., 2022; Xie et al., 2022; Olsson et al., 2022; Von Oswald et al., 2023; Dai et al., 2022b).

Works proposing methods to improve ICL have mainly focused on prompt selection and prompt formulation strategies, Zhou et al. (2023a) propose a different prompting strategy that breaks down a complex problem into a series of simpler subproblems and then solves them in sequence. Zhang et al. (2022b) reformulate the example selection for ICL as a sequential decision problem, and propose a reinforcement learning algorithm for identifying generalizable policies to select demonstration examples. Sorensen et al. (2022) introduces a method for unsupervised selection of prompt templates by maximizing the mutual information between the input and the corresponding model output. Lu et al. (2022) show that the order in which the samples are provided can make a significant difference and propose a method for finding the optimal permutation.

While the majority of approaches to improving ICL have focused on various methods of prompt engineering, such as prompt selection and formulation strategies, our work takes a fundamentally different approach. To the best of our knowledge, this is the first attempt to enhance ICL through a neuron-level analysis, specifically by identifying and pruning neurons that contribute to copying (which we define in Section 3.1) within large language models.

Neuron-Level analysis involves examining individual neurons within the model to determine their specific roles. Relevant studies in this area have aimed to understand and categorize neurons into groups based on their functional roles. For example, Voita et al. (2023) show that individual neu-

rons in LLMs correspond to different groups such as dead neurons, positional neurons, and N-gram neurons. Furthermore, Gurnee et al. (2024) discusses an additional group of categories including entropy neurons, alphabet neurons, syntax neurons, and semantic neurons. Chen et al. (2024) identify safety neurons, which are responsible for safety behaviors. Neuron-level analysis was further expanded to study multilingual LLMs. Tang et al. (2024) detect language-specific and language-agnostic related neurons in multilingual language models. Neuron-Level analysis is typically performed through activation analysis, where one examines the patterns of neuron activations across various inputs (Voita et al., 2023; Gurnee et al., 2024; Stolfo et al., 2024). Attribution methods, such as Integrated Gradients (Sundararajan et al., 2017), are also employed to quantify the contribution of individual neurons to the model's output, thus, allowing the discovery of different neuron familiess (Dai et al., 2022a; Shi et al., 2024).

## 3 METHOD

This section presents our method for detecting and mitigating copying in ICL. First, in section 3.1 we revisit the ICL setting, formally define what are the copying errors, and introduce the Integrated Gradients attribution method. In section 3.2 we elaborate on how we create a synthetic dataset that will be used as a proxy for our proposed detection method, and in section 3.3 we present our method for detecting and mitigating copying neurons.

### 3.1 PRELIMINARIES

**In-Context Learning**  ICL enables a model $f$ to adapt to downstream tasks without any parameter updates. This is achieved by forming a prompt $p$ that includes concatenated training examples. In ICL, a prompt $p$ is constructed by linking the task inputs with their corresponding labels as follows:

$$p = x_1 : y_1, x_2 : y_2, \ldots x_n : y_n, x_{n+1} : \tag{1}$$

Using this prompt, the model $f$ predicts the most probable label $y$ that completes the prefix $p$ according to the function $f$. In this framework, few-shot learning is characterized by the number of examples in the prompt. Furthermore, we denote $S = [y_1, y_2, \ldots y_n]$ as the set of the in-context example answers for prompt $p$, then we say $p$ is an $|S| = n$-shot in-context prompt.

**Copying Bias**  Copying bias, as we define it, refers to the phenomenon where a language model $f$ returns an incorrect answer that is one of the labels $S$ of the in-context samples provided in the prompt. In other words, given a prompt $p$ under the $n$-shot ICL settings, a prediction $y_{n+1}$ is called a copying prediction if $(1) : y_{n+1}$ is a wrong prediction and $(2) : y_{n+1} \in S$.

We hypothesize that there exists a small number of neurons, which we call copying neurons, that trigger the model to copy responses from the prompt examples $S$ of the prompt $p$. The identification of these neurons is, therefore, crucial for understanding how LLMs balance copying and generalization, and for enhancing the reliability and interpretability of these models. We further hypothesize that pruning these neurons by setting their weights to zero would encourage the model to reason rather than solely rely on copying, thereby improving its ability to generalize under few-shot ICL.

**Integrated Gradients (IG)**  Integrated Gradients (IG) is a popular technique in explainable AI (Sundararajan et al., 2017) used to elucidate the relationship between a model's predictions and its input features. It applies to any differentiable model and is computationally efficient. IG works by generating attributions based on integrating the gradients as the input varies between a *baseline* and the final input of interest (the path):

$$\mathbf{IG}(f, \tilde{x}, x, i) = (x_i - \tilde{x}_i) \int_{\alpha=0}^{1} \frac{\partial f(\hat{x}) \cdot d\alpha}{\partial f(\hat{x}_i)} \Bigg|_{\hat{x} = \tilde{x} + \alpha(x - \tilde{x})}, \tag{2}$$

where $f(\cdot)$ denotes the prediction of the model, $x$ is the input vector of interest that we want to attribute, $\tilde{x}$ is the baseline input vector and $i$ is an index denoting the indices of features of interest. The baseline represents a reference point against which the input of interest is compared. More specifically, the baseline is an input vector that is supposed to reflect a neutral, missing, or reference

state. The idea behind using a baseline is to measure how the model's output changes as we transition from this baseline state to the actual input of interest. This change is quantified by integrating the gradients along this path. In our experiments, we use the constant zero baseline as proposed in the original paper of IG. This baseline is straightforward to implement and often provides meaningful attributions. The integral is practically computed using the Riemann sum approximation:

$$\mathbf{IG}(f, \tilde{x}, x, i) \approx \frac{(x_i - \tilde{x}_i)}{m} \sum_{r=1}^{m} \frac{\partial f(\tilde{x} + \frac{r}{m}(x - \tilde{x}))}{\partial x_i}, \tag{3}$$

where $m$ is the total number of Riemann steps.

## 3.2 PROXY ICL DATASET GENERATION

The Integrated Gradients (IG) method operates by backpropagating the probability of the prediction. To effectively utilize IG within our framework, we require a set of in-context examples with ground-truth labels. We avoid relying on access to target task data for neuron selection, as this would limit the practicality of our approach. Instead, we utilize a synthetic proxy dataset to identify and prune neurons on evaluation tasks. This approach also aligns with our research hypothesis that copying neurons are largely task-agnostic. This synthetic dataset is employed within the Integrated Gradients framework to identify copying neurons.

The specific task we choose is that of vowel counting since the mapping from a word to such structural attributes requires reasoning. LLMs can occasionally make errors on this and similar tasks,[1] potentially outputting copying responses. The synthetic samples we utilize simply map an arbitrary word to its corresponding vowel counts (e.g., *apple*: 2). To construct a diverse set of examples, we extract words from a dictionary and calculate their respective vowel counts. In our implementation, we used two examples per prompt to keep the problem tractable while still allowing us to study copying behavior. To eliminate the possibility of label repetition, we ensured that each prompt's examples ($S$) contained distinct vowel counts, with the target word's vowel count always differing from those in $S$. We sampled words with vowel counts ranging from 1 to 8, providing sufficient diversity while avoiding repetition within prompts. For dataset construction, we generated 400 ICL prompts, each using a 2-shot format: 300 examples were used for computing neuron importance scores, and 100 were reserved for validation to determine the optimal pruning configuration. The test answers were carefully designed to never appear in the prompt responses, ensuring clear evaluation of copying behavior.

## 3.3 COPYING NEURONS DETECTION

Denote $V$ as the vocabulary space, $p$ as the in-context prompt of interest containing $n$ in-context examples, and $S_p = [y_1, \dots y_n]$ the set of labels of the different examples in the prompt $p$. In our detection process, we are only interested in prompts $p$ on which the model outputs a wrong prediction $y \in S_p$ (hence the copying) and denote $\hat{y} \notin S_p$ as the ground-truth answer. Let $w^l \in \mathrm{R}^{d_1 \times d_2}$ be the weight matrix of the linear layer at block $l$ on which our detection process operates. Furthermore, we define the model output $\mathrm{P}_p(y|\hat{w}_j^l)$ as the probability of predicting a certain answer $y \in \mathrm{V}$.

$$\mathrm{P}(y|\hat{w}_j^l, p) = \mathrm{P}(y|w_j^l = \hat{w}_j^l, p), \tag{4}$$

where $w_j^{(l)}$ denotes the $j$-th intermediate neuron in the $l$-th layer of interest (Figure 1), $\hat{w}_j^l$ is a given constant that $w_j^l$ is assigned to. We define $l_u = \sum_i^n \mathrm{P}(y = y_i \in S_p|w_j^l = \hat{w}_j^l, p)$ as the probability of predicting an answer which is provided in the prompt examples (in $S$), we also define $l_v = \mathrm{P}(y = \hat{y} \notin S_p|w_j^l = \hat{w}_j^l, p)$ as the probability of predicting the ground-truth answer $\hat{y}$. Lastly, we define $\Delta L = l_v - l_u$ as the prediction shift.

Copying, by definition, occurs when the model's prediction shifts from the true answer to one of the responses provided in the prompt. Thus, copying neurons are those that drive $\Delta L$.

By leveraging IG, we can attribute $\Delta L$ to individual components. This approach enables us to identify specific neurons responsible for copying within the LLM.

---

[1]See, e.g., `https://community.openai.com/t/incorrect-count-of-r-characters-in-the-word-strawberr` `829618`.

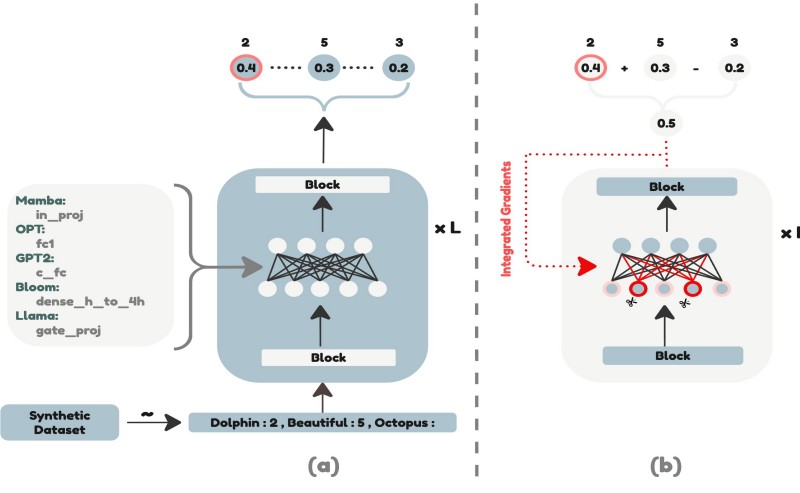

Figure 1: A high-level depiction of our proposed method of detecting copying neurons. First, in (a) we feed ICL prompts from the synthetic dataset. In this phase, we are only interested in the prompts where the model outputs a wrong response which also appears in the prompt examples. Second, in (b) we use these prompts and calculate the sum of the probabilities over predicted responses that appear in the prompt. This sum is used within the IG framework to attribute it to neurons in the targeted layer.

To quantify the contribution of a neuron $w_j^l$ to the prediction shift ($\Delta L$), we gradually change $w_j^{(l)}$ from 0 (the *baseline*) to its original value $\hat{w}_j^{(l)}$ computed by the model and integrate the gradients:

$$\text{Attr}(w_j^l, p, S_p) = \text{IG}(w_j^l) = (\hat{w}_j^l - 0) \int_{\alpha=0}^{1} \frac{\partial |\Delta L|}{\partial w_j^l} d\alpha$$

$$= \sum_i^n \hat{w}_j^{(l)} \int_{\alpha=0}^{1} \frac{\partial \left| \text{P}(y_i | \alpha \hat{w}_j^l, p) - \text{P}(\hat{y} | \alpha \hat{w}_j^l, p) \right|}{\partial w_j^l} d\alpha, \tag{5}$$

$$\text{Attr}(w_j^l, p, S_p) = \text{IG}(w_j^l) = \sum_i^n \frac{\hat{w}_j^l}{m} \sum_{r=1}^{m} \frac{\partial \left| \text{P}(y_i | \frac{r}{m} \hat{w}_j^l, p) - \text{P}(\hat{y} | \frac{r}{m} \hat{w}_j^l, p) \right|}{\partial w_j^l}, \tag{6}$$

where $m = 20$ is the number of approximation steps we use in our experiments, following (Sundararajan et al., 2017).

Finally, we compute the final attribution scores for the neurons $[w_j^l]$ by averaging $\text{Attr}(w_j^l)$ across all samples in the synthetic dataset, resulting in a relevance score that quantifies the extent to which a neuron contributes to copying.

$$\text{R}(w_j^l) = \frac{1}{|D|} \sum_{k=1}^{|D|} \frac{\text{Attr}(w_j^l, p_k, S_{p_k}) - \min_{k'} \text{Attr}(w_j^l, p_{k'}, S_{p_{k'}})}{\max_{k'} \text{Attr}(w_j^l, p_{k'}, S_{p_{k'}}) - \min_{k'} \text{Attr}(w_j^l, p_{k'}, S_{p_{k'}})}, \tag{7}$$

where $D$ is the synthetic dataset and $p_k$ is the $k-$th sample of the synthetic dataset.

To mitigate the copying bias, we propose to suppress the weights of the detected copying neurons as follows:

$$w_i^l = \begin{cases} 0, & \text{R}(w_i^l) \geq \sigma, \\ w_i^l, & \text{R}(w_i^l) < \sigma. \end{cases} \tag{8}$$

where $\sigma$ is the filtering threshold and is tuned using a validation dataset.

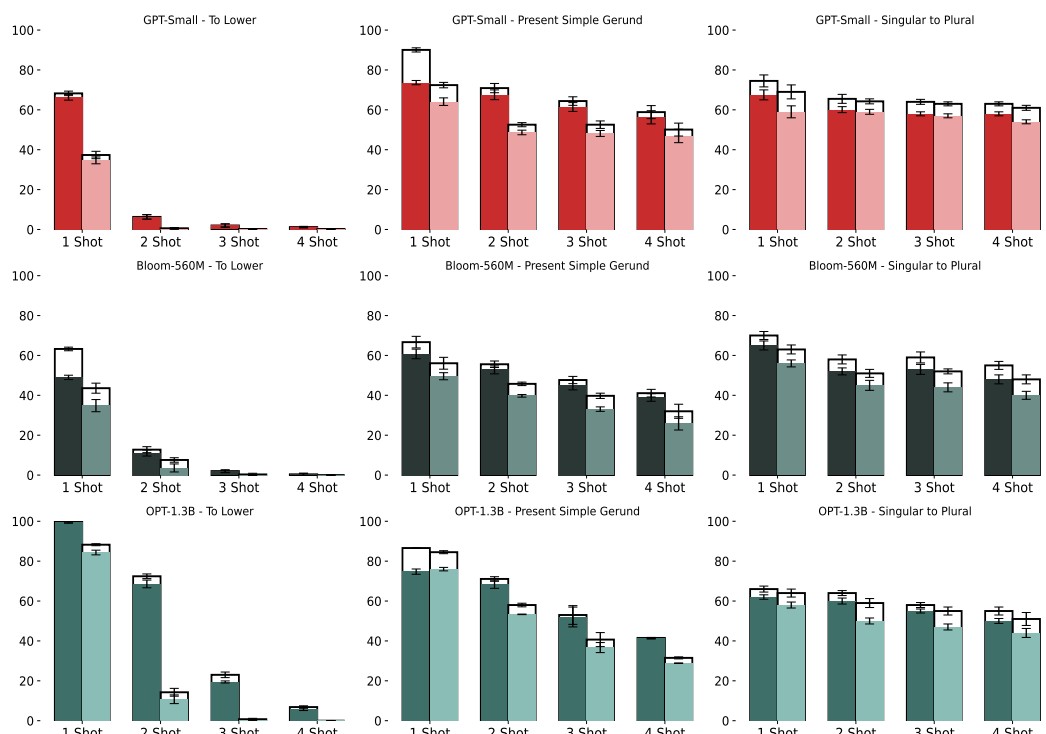

Figure 2: Percentage of total errors and copying errors for both the pruned and un-pruned models, results are shown for 3 ICL tasks across 3 different models: GPT2-Small, BLoom-560M, and OPT-1.3B. The dark bar in each diagram represents the unpruned version while the lighter bar represents the pruned version; the entire bar height represents the total error of the model and the shaded part represents the copying error rate.

## 4 EXPERIMENTS

Our approach offers a generic method for detecting copying neurons, applicable to any language model. To demonstrate the generalizability of our method across different models and tasks, we conduct extensive experiments on a diverse set of LLMs. This includes the recent state-space models, such as Mamba (Gu & Dao, 2023) as well as a broad spectrum of transformer-based models, including OPT (Zhang et al., 2022a), GPT2 (Radford et al., 2019), Bloom (Le Scao et al., 2023) and LLaMA Touvron et al. (2023); Dubey et al. (2024).

**Data** In all of our experiments, we follow Hendel et al. (2023) and Grazzi et al. (2024a) and study 18 tasks in 3 different categories including algorithmic (to lowercase, to uppercase, list first, list last, list max, list min, and next letter), linguistic (present to past, present to gerund, singular to plural, antonyms, and past to perfect), and knowledge (landmark, currency, country to capital, person to language, religion, and continent), the algorithmic tasks are generated automatically, for the linguistic we use the GitHub Repositries[2][3] and the knowledge data is taken from Meng et al. (2022). Additionally, we also incorporate real-world datasets like sentiment classification, including SST2, SST5, and subsets from the BIG-Bench Tasks (Suzgun et al., 2022). For more information about the data, refer to Appendix A.

**Implementation Details** For each model, we use the synthetic validation dataset introduced in Section 3.2 to identify the optimal block and the number of copying neurons to prune as follows. IG, as defined in Equation 7, is applied across the layers of interest (summarized in Appendix. B) in all of the blocks in the model. As described in Section. 3.3, this procedure quantifies which

---

[2]https://github.com/Drulac/English-Verbs-Conjugates
[3]https://github.com/sindresorhus/irregular-plurals

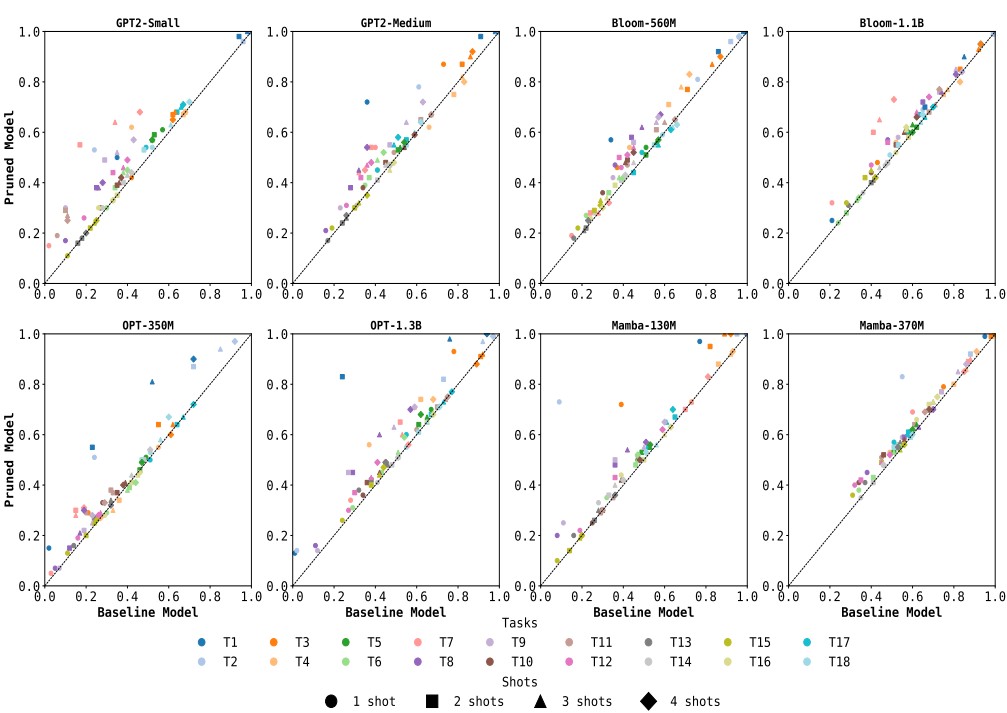

Figure 3: Summary of the results over the synthetic ICL tasks, for more information on the tasks and the exact numbers, refer to Appendix D.

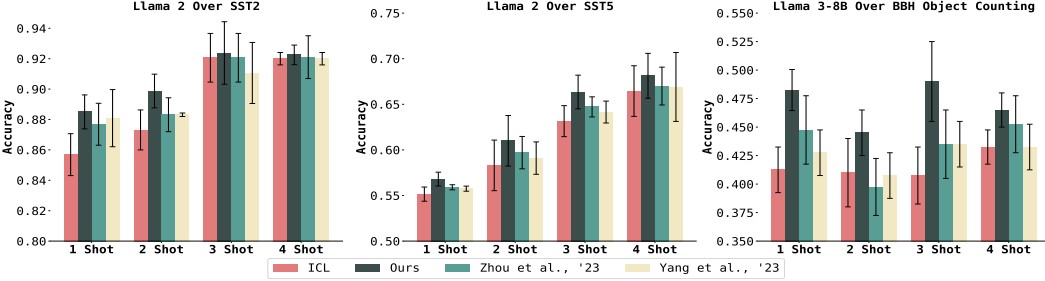

Figure 4: Results of Llama-2 and Llama-3 over SST2, SST5, and Object Counting task from BBH benhmark

neurons contribute most significantly to the copying errors. Furthermore, we use the validation set introduced in Section 3.2 in order to find the optimal pruning rate and the optimal block that maximizes the validation accuracy over the proxy ICL validation set, we apply multiple pruning rates ranging from 1% to 10%, in 1% increments (i.e., [1%, 2%, 3%, . . . , 10%]) over each layer in all of the blocks in the model, and use the best validation accuracy performing configuration to use for the unseen ICL tasks. This procedure allows us to determine the optimal block and the optimal number of neurons to prune for maximizing the accuracy on the validation proxy ICL set. The layers we choose to apply the detection and pruning procedures are summarized in Appendix B.

### 4.1 VALIDATING THE PREVALENCE OF COPYING ERRORS

To validate the significance of copying errors, we first conduct an error analysis for three representative ICL tasks (to lower, singular to plural, and present simple to gerund) across a range of large

language models (GPT2-Small, Bloom-560M, and OPT-1.3B). In this analysis, we show the percentage of copying errors as defined in Section 3.1 out of the total number of errors. The results are presented in Figure 2. In this figure, we present a comparative analysis of error rates across different models and tasks. The dark bars show error rates for the baseline (unpruned) model, while the light bars display results for the pruned model. Each bar is composed of two elements: an unfilled portion representing the total error rate, depicted by the full height of the bar with a black outline, and a filled portion below indicating the copying error rate, which is a subset of the total error rate. Evidently, most of the errors in these few-shot ICL tasks stem from copying, these models tend to replicate responses based on examples provided in the prompt, rather than generalizing to new contexts. Additionally, our pruning method significantly reduces the number of copying errors thus also reducing the total error rate. More analysis on Copying Error can be found in Appendix C.

## 4.2 Tasks Evaluation

To demonstrate the generalizability of our approach across model architectures, we include a range of models: (1) Transformer-based: OPT, GPT-2, BLOOM, LLaMA, and (2) Mamba state space models of various sizes.

For each synthetic ICL task, we utilize the test sets introduced by Hendel et al. (2023). We report the mean accuracy across these test sets for each model and task configuration over the different shots (1, 2, 3, and 4) evaluated using various seeds. Figure. 3 presents a scatter plot comparing the performance of our pruned model against the baseline (non-pruned) model. A diagonal line representing equal performance is included for reference. Data points falling above this line indicate instances where our pruned model achieves higher accuracy than the baseline, while points below the line represent cases where the baseline model outperforms. As evident from the distribution of points that is dominated by those above the diagonal, our pruned model consistently demonstrates superior accuracy across a wide range of ICL tasks for different shot instances, underscoring the effectiveness of our targeting neuron pruning strategy. We believe that the fact our technique rarely leads to a performance drop – and when it does, the impact is only marginal – makes it particularly appealing for practical applications. For the exact numbers, we refer the reader to Appendix D. Additionally, we also test our method on multi-token output tasks, the analysis, and the results are presented in Appendix E. While our pruning method demonstrates substantial improvements in ICL performance, it is important to consider potential tradeoffs in other model capabilities. For a detailed evaluation of these tradeoffs, including the impact on language modeling and knowledge-intensive reasoning tasks, please refer to Appendix F.

In order to test our approach beyond this benchmark ICL tasks, we also test it on tasks that are based on three datasets of collected data: SST-2, SST-5, and the object counting sub-task from the BBH benchmark (Suzgun et al., 2022). In these cases, to allow a comparison with previous work, we use Llama-2 (Touvron et al., 2023) and Llama-3 (Dubey et al., 2024). The same synthetic dataset of vowel mapping is used for copying neurons detection.

We include two recent baselines. The first is Weighted In-Context Learning (WICL) by Yang et al. (2023), which improves the performance of LLMs on downstream tasks by assigning and applying optimal weights to demonstration examples during ICL. The second, Automatic Prompt Engineer (APE) by Zhou et al. (2023b), automatically generates and selects optimal instructions for large language models to improve their performance on various ICL tasks without relying on human-crafted prompts.

The results for these three benchmarks are presented in Figure 4. Evidently, our method significantly improves over the baseline LLM as well as over the two baselines. For a more comprehensive analysis on the effect of shot count beyond the few-shot regime, we conduct additional experiments with up to 64 shots, reported in Appendix G. While our method's improvements are most pronounced in the few-shot setting, it continues to yield consistent benefits even with increased examples.

## 4.3 Tasks-Vector Analysis

Next, we build upon the recent task-vectors framework of Hendel et al. (2023) to study the relationship between copying neurons pruning and the quality of the emerged task vectors in ICL. By comparing the task vectors generated by pruned and unpruned models across various ICL scenarios,

we seek to understand if our proposed targeted pruning can enhance a model's ability to distill task-relevant information from demonstrations, specifically under the few shots settings. Furthermore, we follow the setup of Hendel et al. (2023) and report ICL accuracy using the standard ICL promoting (denoted by ICL), the accuracy obtained by using the emerged Task-Vectors without pruning the copying neurons (denoted as Task-Vectors) and the accuracy obtained by using task vectors obtained from the model with the pruned copying neurons (denoted as Task-Vectors-Pruned).

In Figure. 5, we show the results of OPT-2.7B and Bloom-560M models over the "Singular Plural" and "Country Capital" ICL tasks. As can be seen from the results, pruning the copying neurons indeed yields better Task-Vectors for ICL. This suggests that our pruning strategy may be effectively identifying and removing neurons that were interfering with the model's ability to infer the underlying task. For more results on additional models and tasks, refer to Appendix H .

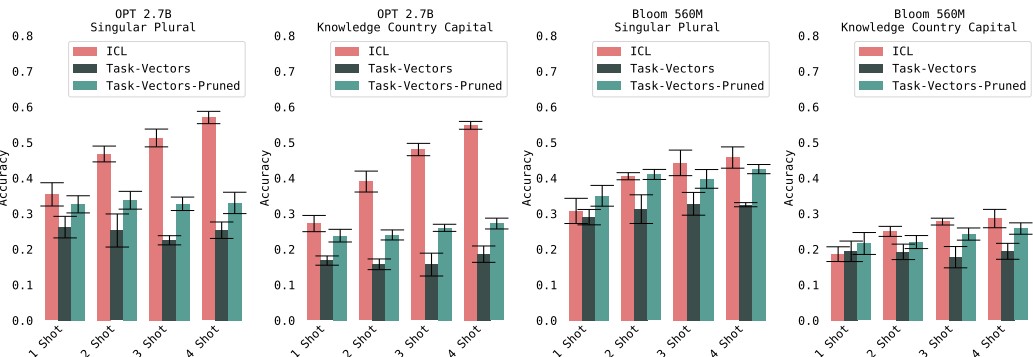

Figure 5: Task-Vectors accuracies over OPT-2.7B and Bloom-560M models tested on (1) Singular Plural and (2) Country Capital ICL tasks. We show the Task-Vectors accuracies with and without pruning the detected copying errors, as can be seen, pruning the copying errors improves the quality of the extracted Task-Vectors across the different shots $\in [1, 2, 3, 4]$ for the two models and ICL tasks.

## 4.4 ABLATION STUDIES

We present multiple ablation studies to evaluate and understand the different components of our proposed detection and pruning methods. These ablation studies were conducted with the OPT-350M, GPT2-Small, and Bloom-560M models, over the Linguistic Antonyms, and Letter to Upper ICL tasks.

Our first experiment focuses on using "Prediction Shift" within the IG framework. We aim to determine whether applying IG to the prediction shift, as defined in Section 3.3, is essential for our proposed method's effectiveness. To this end, we compare our approach with an alternative where IG is applied to the predicted probability (specifically, the maximum probability) instead. Results are presented in Tab.1 ("Max IG" row) clearly shows that the prediction shift is essential for the success of our proposed method.

We further check the effect of min-max normalizing the IG scores across the samples from the proxy ICL task we use in the detection process. The results without this normalization are reported under "w/o Norm". As can be seen, normalizing the scores across the samples can significantly enhance the detection process of the copying neurons.

Additionally, we explore a baseline case where we randomly prune the same percentage of neurons as in the best-performing version of our method. This experiment, labeled as "Random" in Tab.1, shows a degradation in ICL accuracy for some shot settings, underscoring the importance of our targeted pruning strategy. To further evaluate the generality of our copying neuron detection method, we conducted experiments using various proxy ICL tasks beyond vowel counting. The detailed analysis and results can be found in Appendix I. Further analysis of our method's effectiveness across different experimental settings, including combinations with task descriptions, comparisons with fine-tuning approaches, and evaluations on balanced datasets, can be found in Appendix J

| Shot | OPT-350M | | | | | GPT-Small | | | | | Bloom-560M | | | | |
|---|---|---|---|---|---|---|---|---|---|---|---|---|---|---|---|
| | ICL | Ours | Max IG | w/o Norm | Random | ICL | Ours | Max IG | w/o Norm | Random | ICL | Ours | Max IG | w/o Norm | Random |
| **Linguistic Antonyms** | | | | | | | | | | | | | | | |
| 1 | 0.20 | **0.29** | 0.18 | 0.22 | 0.20 | 0.06 | **0.19** | 0.17 | 0.15 | 0.08 | 0.57 | **0.61** | 0.55 | 0.59 | 0.57 |
| 2 | 0.32 | **0.38** | 0.30 | 0.34 | 0.30 | 0.10 | **0.29** | 0.29 | 0.21 | 0.12 | 0.68 | 0.68 | 0.68 | 0.68 | 0.67 |
| 3 | 0.33 | **0.37** | 0.33 | 0.34 | 0.30 | 0.11 | **0.27** | 0.25 | 0.23 | 0.11 | 0.71 | 0.71 | 0.70 | 0.70 | 0.70 |
| 4 | 0.29 | **0.33** | 0.27 | 0.31 | 0.27 | 0.11 | **0.25** | 0.25 | 0.22 | 0.13 | 0.73 | **0.77** | 0.73 | 0.77 | 0.75 |
| **Letter to Uppercase** | | | | | | | | | | | | | | | |
| 1 | 0.24 | **0.51** | 0.30 | 0.37 | 0.23 | 0.24 | **0.53** | 0.37 | 0.45 | 0.34 | 0.76 | **0.81** | 0.77 | 0.78 | 0.76 |
| 2 | 0.72 | **0.87** | 0.72 | 0.81 | 0.72 | 0.96 | 0.96 | 0.95 | 0.96 | 0.96 | 0.92 | **0.96** | 0.91 | 0.95 | 0.90 |
| 3 | 0.85 | **0.94** | 0.85 | 0.90 | 0.84 | 1.00 | 1.00 | 0.98 | 0.99 | 1.00 | 0.96 | **0.98** | 0.96 | 0.96 | 0.96 |
| 4 | 0.92 | **0.97** | 0.91 | 0.94 | 0.92 | 1.00 | 1.00 | 0.99 | 1.00 | 1.00 | 0.96 | **0.98** | 0.95 | 0.95 | 0.95 |

Table 1: Ablation studies for the different components in our method over OPT-350M, GPT2-Small, and Bloom-560M models applied on the Linguistic Antonyms, and Letter to Upper ICL tasks

## 5 CONCLUSIONS AND DISCUSSION

We presented a novel method to mitigate copying bias in few-shot In-Context Learning by pruning neurons that are linked to this behavior according to the integrated gradients interoperability method. Our approach consistently improved performance across a variety of ICL tasks and model architectures. These findings highlight the potential of targeted neuron pruning as an effective strategy for optimizing the capabilities of large language models.

The "out-of-the-box" improvements provided by our method, without the need for task-specific data or fine-tuning, have significant practical implications for deploying more reliable few-shot learning systems. Our approach allows for the enhancement of LLM performance across a wide range of tasks using only a simple, synthetic dataset for neuron identification. Moreover, the consistent improvements observed across different model architectures suggest that this method could be broadly applicable, potentially becoming a standard post-processing step in LLM deployment pipelines.

The success of our pruning method in improving performance across various tasks indicates that "copying neurons" may be acting as a form of shortcut, inhibiting the model's ability to engage in more sophisticated reasoning processes. This observation aligns with recent work on shortcut learning in neural networks (Yom Din et al., 2024; Belrose et al., 2023) and suggests that in-context learning quality could potentially be improved by carefully modulating the influence of different neuron groups or pathways.

Our results suggest that by pruning copying neurons, we enhance the model's ability to distill task-relevant information from demonstrations, leading to more effective task vectors. This raises interesting questions about the relationship between neuron-level representations and the higher-level task embeddings captured by task vectors. Specifically, it may be useful to consider the representation in a way that disentangles multiple activation pathways.

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

# A    TASKS DATASETS

In all of our experiments, we follow Hendel et al. (2023) and Grazzi et al. (2024a) and study 18 tasks in 4 different categories including algorithmic, translation, linguistic, and knowledge, the algorithmic tasks are generated automatically, for the linguistic we use the GitHub Repositories[4][5] and the knowledge data is taken from Meng et al. (2022). More details on the datasets are shown in Table A.

| Category | Reference | Task | Example |
|----------|-----------|------|---------|
| Algorithmic | T1 | To Lowercase | A → a |
| | T2 | To Uppercase | a → A |
| | T3 | List First | q,b,e,s → q |
| | T4 | List Last | q,b,e,s → s |
| | T5 | List Max | 2,1,5 → 5 |
| | T6 | List Min | 2,1,5 → 1 |
| | T7 | Next Letter | a,b,c → d |
| Linguistic | T8 | Present to past | go → went |
| | T9 | Present to gerund | go → going |
| | T10 | Singular to plural | cat → cats |
| | T11 | Antonyms | happy → sad |
| | T12 | Past to Perfect | catch → caught |
| Knowledge | T13 | Landmark | Maybach → Germany |
| | T14 | Currency | Azerbaijan → Manat |
| | T15 | Country to Capital | France → Paris |
| | T16 | Person to Language | Macron → French |
| | T17 | Religion | Muhammad → Islam |
| | T18 | Continent | Swanson Mountains → Antarctica |

Beyond synthetic ICL tasks, we use sentiment classification datasets like SST2, and SST5, in SST2 the task is to classify text sentences into one of the two sentiments (negative or positive), while in SST5 the task is to classify text sentences into one of five sentiments (very positive, positive, neutral, negative, very negative). Additionally, we also incorporate the object-counting task from the BBH benchmark (Suzgun et al., 2022), where the task is to find out the total number of objects given in a context sentence, a sample illustration from the dataset is as follows:

"I have a car, and a toaster. How many objects do I have? → 2"

---

[4]https://github.com/Drulac/English-Verbs-Conjugates
[5]https://github.com/sindresorhus/irregular-plurals

## B    TARGET LAYERS

This section outlines the exact layers targeted by our detection and pruning techniques. We concentrate on specific linear layers within each model block, with GPT2 being an exception where we focus on a CNN layer. Our approach encompasses both transformer-based and Mamba-based architectural designs. For a detailed breakdown of the exact layers our method operates on across various model families, refer to Table 2. The hyperparameters used in the pruning process including target layer and the percentage of neurons to prune are summarized in Table 3.

Table 2: A summary of the specific layers on which we apply our detection and pruning method for different model families

| Family | Layer | type |
|--------|-------|------|
| OPT | fc1 | linear |
| GPT2 | mlp.c_fc | cnn |
| Bloom | mlp.dense_h_to_4h | linear |
| Llama | mlp.gate_proj | linear |
| Mamba | mixer.in_proj | linear |

Table 3: A summary of the meta-optimization procedure

| Family | Pruned Layer | Percentage |
|--------|--------------|------------|
| OPT 125M | 5 | 5% |
| OPT 350M | 0 | 5% |
| OPT 1.3B | 14 | 7% |
| OPT 2.7B | 19 | 3% |
| GPT2-Small | 0 | 5% |
| GPT2-Medium | 6 | 7% |
| Bloom 560M | 11 | 8% |
| Bloom 1.1B | 0 | 5% |
| Mamba 130M | 0 | 8% |
| Mamba 370M | 0 | 6% |

# C    ADDITIONAL ERROR ANALYSIS RESULTS

We believe copying errors dominate because it's a "safer" failure mode for the model - repeating a previously seen answer rather than generating a potentially incorrect novel response. This aligns with the training objective of minimizing unlikely outputs. For the vowel-counting task, copying errors occur at a similar rate to other tasks, suggesting that copying behavior is a general phenomenon rather than task-specific. This consistency between in-distribution and out-of-distribution tasks supports our hypothesis that we're identifying fundamental copying mechanisms in the model architecture rather than task-specific patterns.

Table 4 reports the ratio of copying errors to total errors (Copying Error Rate / Total Error Rate) over the vowel counting task for a single run, demonstrating the prevalence of copying behavior across model scales and architectures. To validate our method's effectiveness in mitigating copying errors,

Table 4: Ratio of copying errors to total errors on the vowel counting task.

| Model | 1 Shot | 2 Shot | 3 Shot | 4 Shot |
|---|---|---|---|---|
| GPT2-Small | 0.936 | 0.566 | 0.669 | 0.733 |
| GPT2-Medium | 0.853 | 0.430 | 0.711 | 0.812 |
| Bloom 560M | 0.751 | 0.390 | 0.580 | 0.686 |
| Bloom 1.1B | 0.987 | 0.680 | 0.777 | 0.906 |
| OPT 125M | 0.802 | 0.657 | 0.689 | 0.680 |
| OPT 350M | 0.227 | 0.775 | 0.812 | 0.872 |
| OPT 1.3B | 0.951 | 0.378 | 0.682 | 0.800 |
| OPT 2.7B | 0.942 | 0.439 | 0.621 | 0.712 |
| Llama3 8B | 0.835 | 0.500 | 0.691 | 0.792 |

we analyze the percentage of copying errors among total errors on the BBH ICL task before and after applying our pruning method. Table 5 shows that our approach substantially reduces copying errors even in larger models like Llama3 8B, with reductions of up to 60% in some settings.

Table 5: Percentage of copying errors in total errors on the BBH ICL task.

| Model | 1 Shot | 2 Shot | 3 Shot | 4 Shot |
|---|---|---|---|---|
| Llama3 8B ICL | 10% | 16% | 19% | 24% |
| Llama3 8B ICL + Ours | **4%** | **10%** | **12%** | **15%** |

## D  MORE RESULTS

This section presents the full results of the ICL tasks. We compare the performance of various models across 18 ICL tasks. Each entry in the result tables contains two values: the left value represents the performance of the unpruned model, while the right value shows the performance achieved using our proposed method. Results are averaged across 5 different seeds.

Table 6: Results over the GPT2 model family. The left number is the base model and the right number is with our pruning approach. Results are averaged across 5 different runs.

| Task | GPT2-Small | | | | GPT2-Medium | | | |
|------|--------|--------|--------|--------|--------|--------|--------|--------|
|      | 1-shot | 2-shot | 3-shot | 4-shot | 1-shot | 2-shot | 3-shot | 4-shot |
| T1  | .35\|**.50** | .94\|**.98** | .98\|**1.0** | .99\|**1.0** | .36\|**.72** | .91\|**.98** | .98\|**1.0** | .99\|**1.0** |
| T2  | .24\|**.53** | .96\|.96 | 1.0\|1.0 | 1.0\|1.0 | .61\|**.78** | .99\|**1.0** | .99\|**1.0** | 1.0\|1.0 |
| T3  | .42\|.42 | .62\|**.67** | .63\|**.68** | .62\|**.65** | .73\|**.87** | .82\|**.87** | .86\|**.90** | .87\|**.92** |
| T4  | .42\|**.62** | .67\|.67 | .68\|.68 | .68\|.68 | **.66**\|.62 | **.78**\|.75 | .82\|.82 | **.83**\|.80 |
| T5  | .57\|**.61** | .53\|**.59** | .52\|**.57** | .52\|**.57** | .54\|.54 | .51\|**.53** | .52\|**.54** | .54\|**.56** |
| T6  | .30\|.30 | .34\|**.38** | .38\|**.44** | .40\|**.45** | .35\|**.39** | .37\|**.42** | .41\|**.49** | .44\|**.52** |
| T7  | .02\|**.15** | .17\|**.55** | .34\|**.64** | .46\|**.68** | .40\|**.54** | .38\|**.54** | .32\|**.44** | .35\|**.45** |
| T8  | .10\|**.17** | .25\|**.38** | .26\|**.38** | .28\|**.40** | .16\|**.21** | .28\|**.38** | .32\|**.45** | .36\|54 |
| T9  | .10\|**.30** | .29\|**.49** | .35\|**.52** | .43\|**.57** | .23\|**.30** | .47\|**.55** | .55\|**.64** | .63\|**.72** |
| T10 | .27\|**.30** | .35\|**.39** | .36\|**.40** | .37\|**.42** | .34\|**.38** | .45\|**.48** | .54\|.54 | .59\|.59 |
| T11 | .06\|**.19** | .10\|**.29** | .11\|**.27** | .11\|**.25** | .49\|**.52** | .62\|**.65** | .67\|.67 | .67\|.67 |
| T12 | .19\|**.26** | .33\|**.44** | .38\|46 | .40\|**.49** | .26\|**.31** | .33\|**.42** | .36\|**.47** | .38\|48 |
| T13 | .18\|.18 | .16\|.16 | .18\|.18 | .20\|.20 | .17\|.17 | .24\|.24 | .26\|.26 | .26\|**.27** |
| T14 | .28\|**.30** | .38\|**.40** | .40\|**.43** | .42\|**.44** | .31\|.31 | .41\|.41 | .45\|**.47** | .47\|.47 |
| T15 | .11\|.11 | .22\|.22 | .24\|.24 | .25\|.25 | .19\|**.22** | .30\|.30 | .32\|.32 | **.36**\|.35 |
| T16 | .26\|**.30** | .35\|.35 | .33\|.33 | .33\|.33 | .41\|**.45** | .47\|.47 | **.47**\|.45 | **.49**\|.48 |
| T17 | .49\|**.54** | .64\|**.68** | .66\|**.70** | .67\|**.71** | .41\|**.45** | .55\|**.57** | .49\|**.55** | .51\|**.58** |
| T18 | .52\|**.54** | .48\|**.53** | .61\|**.63** | .70\|**.72** | .54\|.54 | .55\|.55 | .59\|**.60** | .62\|**.64** |

Table 7: Results over the BLOOM models family. The left number is the base model and the right number is with our pruning approach. Results are averaged across 5 different runs.

| Task | Bloom-560M | | | | Bloom-1.1B | | | |
|------|--------|--------|--------|--------|--------|--------|--------|--------|
|      | 1-shot | 2-shot | 3-shot | 4-shot | 1-shot | 2-shot | 3-shot | 4-shot |
| T1  | .34\|**.57** | .86\|**.92** | .98\|**1.0** | .99\|**1.0** | .21\|**.25** | .66\|**.70** | .85\|**.90** | .93\|**.95** |
| T2  | .76\|**.81** | .92\|**.96** | .96\|**.98** | .96\|**.98** | .65\|**.69** | .99\|.99 | 1.0\|1.0 | 1.0\|1.0 |
| T3  | .37\|**.46** | .71\|**.77** | .83\|**.87** | .87\|**.90** | .43\|**.48** | .83\|**.85** | .92\|**.93** | .93\|**.95** |
| T4  | .43\|**.54** | .62\|**.71** | .68\|**.78** | .72\|**.83** | .52\|**.56** | .75\|.75 | .77\|.77 | **.83**\|.80 |
| T5  | .51\|**.54** | .51\|.51 | .56\|.56 | .57\|.57 | .58\|**.60** | .62\|.62 | .60\|**.63** | .60\|.60 |
| T6  | .22\|**.27** | .33\|**.36** | .38\|**.42** | .40\|**.42** | .24\|.24 | .28\|.28 | .33\|.33 | .34\|.34 |
| T7  | .15\|**.19** | .24\|**.28** | .28\|.28 | **.33**\|.32 | .21\|**.32** | .41\|**.60** | .44\|**.65** | .51\|**.73** |
| T8  | .36\|**.47** | .44\|**.58** | .49\|**.62** | .58\|**.67** | .48\|**.57** | .65\|**.72** | .74\|**.76** | .81\|**.83** |
| T9  | .35\|**.42** | .45\|**.56** | .56\|**.64** | .57\|**.66** | .53\|**.58** | .73\|**.76** | .81\|**.85** | .84\|.84 |
| T10 | .30\|**.36** | .42\|**.49** | .41\|**.48** | .45\|**.52** | .40\|**.44** | .52\|**.55** | .56\|**.60** | .62\|**.66** |
| T11 | .42\|**.47** | .56\|**.61** | .60\|**.64** | .65\|.65 | .57\|**.61** | .68\|.68 | .71\|.71 | .73\|**.77** |
| T12 | .39\|**.46** | .38\|**.50** | .45\|**.54** | .42\|**.51** | .40\|**.47** | .48\|**.56** | .62\|**.68** | .68\|**74** |
| T13 | .16\|**.18** | .22\|.22 | .21\|.21 | .23\|**.25** | .29\|**.31** | .40\|.40 | .41\|.41 | .42\|.42 |
| T14 | .26\|**.28** | .39\|**.41** | .45\|**.48** | .41\|**.43** | .36\|.36 | .48\|.48 | .44\|**.46** | .47\|.47 |
| T15 | .18\|**.22** | .26\|**.29** | .29\|**.33** | .29\|**.31** | .28\|**.32** | .37\|**.42** | .40\|**.45** | .42\|.42 |
| T16 | .24\|**.26** | .36\|**.39** | .32\|**.34** | .30\|.30 | .52\|.52 | .58\|.58 | .57\|**.61** | .57\|**.62** |
| T17 | .49\|**.52** | **.45**\|.44 | **.57**\|.55 | **.63**\|.61 | .57\|**.60** | .63\|**.67** | .66\|.66 | .70\|.70 |
| T18 | .59\|.59 | .55\|.55 | **.64**\|.62 | **.66**\|.63 | .49\|**.51** | .53\|**.55** | .64\|.64 | .65\|**.67** |

Table 8: Results over the OPT models family, the left number is the base model and the right number is with our pruning approach, results are averaged across 5 different runs

|  | OPT-125M | | | | OPT-350M | | | |
|---|---|---|---|---|---|---|---|---|
| **Task** | **1 Shot** | **2 Shot** | **3 Shot** | **4 Shot** | **1 Shot** | **2 Shot** | **3 Shot** | **4 Shot** |
| T1 | .01\|**.18** | .37\|**.59** | .21\|**.35** | .21\|**.31** | .02\|**.15** | .23\|**.55** | .52\|**.81** | .72\|**.90** |
| T2 | .02\|**.17** | .27\|**.49** | .25\|**.28** | .21\|**.24** | .24\|**.51** | .72\|**.87** | .85\|**.94** | .92\|**.97** |
| T3 | .19\|**.34** | .30\|**.38** | .32\|**.38** | .32\|**.39** | .21\|**.29** | .55\|**.64** | .62\|**.64** | **.61**\|.60 |
| T4 | .18\|**.37** | .30\|**.40** | .30\|**.40** | .29\|**.39** | .55\|.55 | **.36**\|.34 | **.33**\|.30 | .27\|.27 |
| T5 | .32\|**.39** | .33\|.33 | **.30**\|.29 | .30\|.30 | .49\|**.51** | .46\|.46 | .46\|.46 | .47\|**.49** |
| T6 | .25\|**.35** | .31\|**.35** | .30\|**.34** | .33\|**.35** | **.30**\|.29 | **.41**\|.39 | **.40**\|.38 | **.44**\|.41 |
| T7 | .01\|**.07** | .03\|**.12** | .03\|**.09** | .04\|**.08** | .03\|**.05** | .15\|**.30** | .15\|**.28** | .19\|**.31** |
| T8 | .01\|.01 | .03\|**.05** | .04\|**.07** | .04\|**.06** | .05\|**.07** | .12\|**.15** | .17\|**.21** | .19\|**.30** |
| T9 | .01\|**.06** | .11\|**.16** | .14\|**.18** | .15\|**.21** | .07\|.07 | .19\|**.22** | .23\|**.25** | .23\|**.28** |
| T10 | .10\|**.19** | .22\|**.25** | .28\|**.30** | .29\|**.32** | .28\|**.33** | .35\|**.37** | .39\|**.41** | .38\|**.40** |
| T11 | .05\|**.10** | .11\|**.15** | .15\|**.17** | .16\|**.18** | .20\|**.29** | .32\|**.38** | .33\|**.37** | .29\|**.33** |
| T12 | .02\|**.09** | .12\|**.16** | .16\|.16 | .16\|.16 | .16\|**.19** | .24\|**.26** | .27\|**.29** | .26\|**.28** |
| T13 | .08\|**.12** | .07\|**.12** | .10\|**.15** | .11\|**.16** | .14\|**.16** | .24\|**.27** | .32\|**.34** | .32\|.32 |
| T14 | .09\|**.25** | .26\|**.34** | .31\|**.33** | .31\|**.34** | .33\|**.37** | .48\|**.50** | .42\|**.44** | .51\|**.54** |
| T15 | .04\|**.09** | .13\|**.16** | .15\|**.18** | .15\|**.17** | .11\|**.13** | .20\|.20 | .24\|**.25** | .25\|**.26** |
| T16 | .19\|**.24** | .26\|**.30** | .25\|**.28** | .24\|**.26** | **.29**\|.28 | **.42**\|.41 | **.45**\|.44 | **.46**\|.45 |
| T17 | .54\|.54 | .54\|**.57** | .53\|**.55** | .62\|.62 | **.51**\|.50 | .64\|.64 | .67\|.67 | .72\|.72 |
| T18 | .56\|.56 | **.65**\|.63 | .65\|.65 | .69\|.69 | .51\|**.53** | .47\|**.50** | .56\|**.58** | .60\|**.67** |

|  | OPT-1.3B | | | | OPT-2.7B | | | |
|---|---|---|---|---|---|---|---|---|
| **Task** | **1 Shot** | **2 Shot** | **3 Shot** | **4 Shot** | **1 Shot** | **2 Shot** | **3 Shot** | **4 Shot** |
| T1 | .01\|**.13** | .24\|**.83** | .76\|**.98** | .94\|**1.0** | .09\|**.13** | .89\|**.98** | .99\|**1.0** | 1.0\|1.0 |
| T2 | .02\|**.14** | .73\|**.82** | .92\|**.97** | .97\|**.99** | .04\|**.12** | .72\|**.93** | .86\|**.98** | .91\|**1.0** |
| T3 | .78\|**.93** | .91\|.91 | .92\|.92 | **.89**\|.88 | .84\|**.90** | .93\|**.95** | .97\|**1.0** | .98\|**1.0** |
| T4 | .37\|**.56** | .62\|**.74** | .59\|**.71** | .68\|**.74** | .47\|**.57** | .64\|**.76** | .77\|**.80** | .74\|**.76** |
| T5 | .67\|**.70** | .61\|**.64** | .65\|**.67** | .62\|**.68** | .56\|.56 | .49\|.49 | .54\|.54 | .55\|.55 |
| T6 | .29\|**.31** | .43\|**.45** | .51\|**.53** | .54\|**.59** | .35\|**.37** | .47\|.47 | .51\|**.52** | .54\|.54 |
| T7 | .28\|**.34** | .52\|**.65** | .53\|**.60** | .56\|.56 | .14\|**.23** | .44\|**.50** | .50\|**.58** | .44\|**.49** |
| T8 | .11\|**.16** | .29\|**.45** | .42\|**.60** | .57\|**.70** | .20\|**.32** | .49\|**.65** | .61\|**.73** | .68\|**.75** |
| T9 | .12\|**.14** | .27\|**.45** | .49\|**.63** | .59\|**.71** | .28\|**.39** | .68\|**.75** | .80\|**.83** | .82\|**.85** |
| T10 | .34\|**.36** | .36\|**.41** | .42\|**.45** | .45\|**.49** | .35\|**.40** | .46\|**.50** | .52\|**.56** | .59\|**.62** |
| T11 | .60\|**.62** | .70\|**.71** | .74\|.74 | .75\|.75 | .64\|.64 | .75\|**.77** | .76\|**.78** | .78\|.78 |
| T12 | .27\|**.30** | .30\|**.37** | .38\|**.43** | .41\|**.49** | .28\|**.33** | .40\|**.47** | .42\|**.44** | .47\|**.52** |
| T13 | .32\|**.38** | .38\|**.42** | .46\|**.48** | .45\|**.49** | .42\|**.46** | .50\|**.52** | .55\|**.57** | .61\|.61 |
| T14 | .41\|.41 | .48\|.48 | .51\|.51 | .51\|.51 | .43\|.43 | .53\|**.55** | .49\|**.50** | .52\|.52 |
| T15 | .24\|**.26** | .38\|**.40** | .42\|**.44** | .44\|**.47** | .25\|**.30** | .40\|**.45** | .50\|**.54** | .53\|**.57** |
| T16 | .56\|.56 | .65\|.65 | .65\|.65 | .68\|.68 | .70\|.70 | .74\|.74 | .72\|.72 | .74\|.74 |
| T17 | .55\|**.60** | .67\|**.68** | .73\|**.73** | .77\|.77 | .54\|**.59** | .62\|**.64** | .72\|.72 | .72\|.72 |
| T18 | .55\|.55 | .61\|.61 | .65\|.65 | .71\|.71 | .48\|**.50** | .59\|.59 | .69\|.69 | .73\|.73 |

Table 9: Results over the Mamba models family. The left number is the base model and the right number is with our pruning approach. Results are averaged across 5 different runs.

| Task | Mamba-130M | | | | Mamba-370M | | | |
|---|---|---|---|---|---|---|---|---|
| | 1-shot | 2-shot | 3-shot | 4-shot | 1-shot | 2-shot | 3-shot | 4-shot |
| T1 | .77\|**.97** | 1.0\|1.0 | 1.0\|1.0 | 1.0\|1.0 | .95\|**.99** | 1.0\|1.0 | 1.0\|1.0 | 1.0\|1.0 |
| T2 | .09\|**.73** | .95\|**1.0** | 1.0\|1.0 | 1.0\|1.0 | .55\|**.83** | .88\|**.92** | 1.0\|1.0 | 1.0\|1.0 |
| T3 | .39\|**.72** | .82\|**.95** | .89\|**1.0** | .92\|**1.0** | .75\|**.79** | .98\|**.99** | .99\|**1.0** | 1.0\|1.0 |
| T4 | .73\|.73 | .86\|**.88** | .92\|.92 | .93\|.93 | .49\|**.53** | .80\|.80 | .86\|.86 | 0.91\|**.93** |
| T5 | .47\|**.50** | .49\|**.53** | .53\|**.55** | .53\|**.56** | .53\|**.56** | .51\|**.53** | .62\|**.64** | .60\|**.62** |
| T6 | .32\|**.35** | .46\|**.48** | .47\|**.50** | .47\|**.52** | .34\|**.38** | .41\|**.43** | .51\|**.53** | .54\|**.55** |
| T7 | .14\|.14 | .70\|.70 | .73\|.73 | .81\|**.83** | .60\|**.69** | .87\|**.89** | .88\|**.90** | .85\|.85 |
| T8 | .08\|**.20** | .36\|**.48** | .42\|**.54** | .51\|**.57** | .38\|45 | .56\|**.59** | .63\|.63 | .70\|.70 |
| T9 | .11\|**.25** | .36\|**.50** | .50\|**.55** | .60\|**.65** | .55\|**.59** | .74\|**.77** | .82\|**.85** | .86\|**.88** |
| T10 | .25\|.25 | .30\|.30 | .39\|**.42** | .48\|**.50** | .34\|**.41** | .46\|**.52** | .56\|**.58** | .68\|**.70** |
| T11 | .20\|.20 | .29\|.29 | .35\|.35 | .30\|.30 | .45\|**.51** | .61\|**.64** | .70\|**.73** | .66\|**.69** |
| T12 | .19\|**.22** | .36\|**.43** | .47\|**.52** | .59\|**.62** | .32\|**.40** | .35\|**.42** | .45\|**.49** | .49\|**.52** |
| T13 | .16\|**.20** | .26\|.26 | .28\|**.30** | .36\|.36 | .37\|**.41** | .52\|**.55** | .54\|**.57** | .53\|**.55** |
| T14 | .28\|**.33** | .33\|**.36** | .36\|**.40** | .40\|**.42** | .35\|.35 | .41\|.41 | .46\|**.48** | .46\|**.48** |
| T15 | .08\|**.10** | .14\|.14 | .19\|.19 | .20\|.20 | .31\|**.36** | .45\|**.49** | .54\|.54 | .56\|.56 |
| T16 | .39\|**.44** | .50\|.50 | .60\|.60 | .63\|.63 | .62\|**.66** | .68\|**.72** | .70\|**.73** | .72\|**.75** |
| T17 | .52\|**.55** | .65\|**.67** | .61\|**.65** | .64\|**.70** | .51\|**.57** | .58\|**.61** | .58\|**.61** | .58\|**.60** |
| T18 | .48\|**.50** | .48\|**.50** | .56\|.56 | .51\|**.53** | .53\|**.56** | .51\|**.55** | .59\|.59 | .60\|.60 |

# E    MULTI TOKEN OUTPUTS

To validate our method's effectiveness on multi-token outputs, we evaluate performance on two tasks: country-currency pairs (e.g., "Benin" → "CFA Franc (XOF)") and national parks (e.g., "Badlands National Park" → "South Dakota"). Despite using single-token vowel counting as our proxy task, we observe consistent improvements across both tasks for Bloom 560M and GPT2-Small models in all shot settings Table 10 and Table 11 (results are averaged across 5 different seeds, with the left number showing baseline performance and the right number showing our method's results. ). While our method could be extended to explicitly handle multi-token outputs by aggregating prediction shifts across the sequence, our results suggest this is unnecessary as the current approach already generalizes effectively to multi-token scenarios.

Table 10: Results on National Parks task

| Model | 1 Shot | 2 Shot | 3 Shot | 4 Shot |
|---|---|---|---|---|
| Bloom 560M | 0.03\|**0.12** | 0.06\|**0.15** | 0.075\|**0.19** | 0.07\|**0.20** |
| GPT2-Small | 0.04\|**0.09** | 0.09\|**0.14** | 0.10\|**0.12** | 0.10\|**0.14** |

Table 11: Results on Country-Currency task

| Model | 1 Shot | 2 Shot | 3 Shot | 4 Shot |
|---|---|---|---|---|
| Bloom 560M | 0.06\|**0.10** | 0.12\|**0.15** | 0.12\|**0.17** | 0.11\|**0.15** |
| GPT2-Small | 0.06\|**0.10** | 0.08\|**0.12** | 0.09\|**0.12** | 0.09\|**0.13** |

# F    TRADEOFFS

## F.1    MMLU AND PERPLEXITY

While our pruning method significantly improves ICL performance, it is crucial to understand potential tradeoffs in other capabilities of the model. To assess this, we evaluated the pruned models on two fundamental downstream tasks: (1) base language modeling capability through perplexity on WikiText, and (2) knowledge-intensive reasoning through the MMLU benchmark (Hendrycks et al., 2020). As shown in Table 12, pruning the copying neurons results in a minimal degradation in perplexity across all model sizes and architectures, with the relative increase ranging from 0.008% (OPT-2.7B) to 12.25% (Llama3-8B). Most models show less than 2% degradation, suggesting that our targeted neuron pruning primarily affects copying circuits while largely preserving general language modeling capabilities. For knowledge-intensive reasoning, Table 13 shows that performance on MMLU's diverse categories (humanities, social sciences, STEM, and other) remains stable or even improves in certain cases. For instance, GPT2-Small shows improved performance in social sciences (+2.11%) and STEM (+0.37%), while Bloom-1.1B demonstrates gains across humanities (+0.20%), STEM (+0.16%), and other categories (+0.25%). Notably, larger models like Llama3-8B maintain their strong performance across all categories after pruning, with negligible changes in accuracy (within ±0.3%). The minimal impact on both perplexity and MMLU performance, combined with the substantial improvements in ICL performance demonstrated, indicates a favorable tradeoff.

Table 12: Perplexity on WikiText test (lower is better)

| Model | Without Pruning | With Pruning |
|---|---|---|
| GPT-Small | **25.188** | 25.548 |
| GPT-Medium | **18.473** | 18.590 |
| Bloom 560M | **21.328** | 21.397 |
| Bloom 1.1B | **16.836** | 16.883 |
| OPT 125M | **26.119** | 26.432 |
| OPT 350M | **20.888** | 20.940 |
| OPT 1.3B | **13.889** | 13.904 |
| OPT 2.7B | **11.757** | 11.758 |
| Llama2 7B | **6.542** | 6.549 |
| Llama3 8B | **5.184** | 5.819 |

Table 13: MMLU Test Performance (accuracy)

| Model | Humanities | Social Sciences | STEM | Other |
|---|---|---|---|---|
| GPT2-Small | 0.2421 | 0.2171 | 0.2131 | **0.2382** |
| GPT2-Small + Ours | 0.2421 | **0.2382** | **0.2168** | 0.2128 |
| GPT2-Medium | 0.2290 | **0.2427** | **0.2350** | 0.2128 |
| GPT2-Medium + Ours | **0.2427** | 0.2379 | 0.2294 | **0.2144** |
| Bloom-560M | 0.2294 | **0.2419** | **0.2391** | 0.2138 |
| Bloom-560M + Ours | **0.2421** | 0.2398 | 0.2164 | **0.2141** |
| Bloom-1.1B | 0.2482 | **0.2626** | 0.2252 | 0.2347 |
| Bloom-1.1B + Ours | **0.2502** | 0.2620 | **0.2268** | **0.2372** |
| OPT-2.7B | **0.2670** | 0.2398 | **0.2470** | **0.2688** |
| OPT-2.7B + Ours | 0.2648 | **0.2427** | 0.2457 | 0.2677 |
| Llama2-7B | **0.4329** | 0.5484 | 0.5297 | **0.3606** |
| Llama2-7B + Ours | 0.4327 | **0.5491** | 0.5297 | 0.3603 |
| Llama3-8B | **0.5501** | **0.7078** | 0.7322 | 0.5373 |
| Llama3-8B + Ours | 0.5486 | 0.7048 | **0.7329** | **0.5375** |

## F.2  NEEDLE IN A HAYSTACK

To verify our pruning method preserves essential model capabilities, we evaluated performance on the needle-in-haystack benchmark[6] over Llama2 7B model. This benchmark tests a model's ability to locate and extract specific information from long contexts. In our experiments, we used PaulGrahamEssays as the context 'high-stack' and embedded the sentence "The best thing do in San Francisco is eat a sandwich and sit in Dolores Park on a sunny day" as the 'needle'. Our experiments in Figure 6 show slight differences in performance between pruned and unpruned versions of the tested model across varying context lengths. The high consistency across context and depth length demonstrates that while our pruning effectively mitigates copying bias in ICL settings, it does not compromise any model's fundamental ability to process and recall information from long contexts.

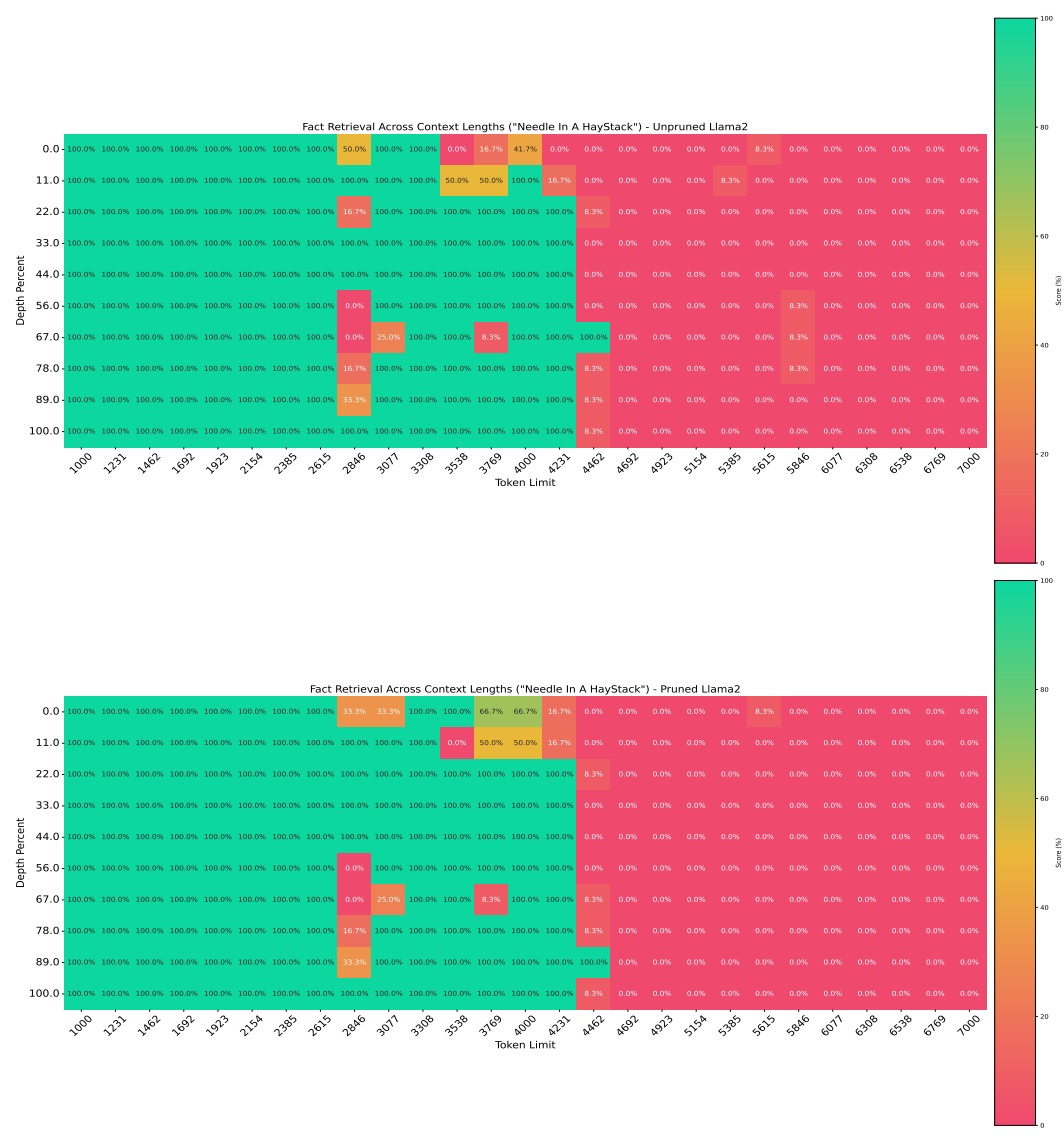

Figure 6: Needle in A Haystack evaluation over Llama2 7B model. The upper plot shows the results for the unpruned model while the lower plot shows the results for the pruned model.

---

[6]https://github.com/gkamradt/LLMTest_NeedleInAHaystack

## F.3 EVALUATION ON ALPACAEVAL2

We tested our approach on AlpacaEval2[7]. Due to computational constraints, we evaluated on half of the test set. For reference, the original Llama2's performance on the full test set achieves 5.4% LC win rate and 5.0% win rate. Results are shown in Table 14. On our half-set evaluation, the unpruned Llama2 achieves 4.84% LC win rate and 2.69% win rate, while our pruned version improves to 5.26% LC win rate and 3.21% win rate. This (along with results on MMLU) suggests that our pruning method preserves the model's ability to handle both general language tasks.

Table 14: Performance comparison on AlpacaEval2 (half test set)

| Model | LC Win Rate | Win Rate |
|---|---|---|
| Unpruned Llama2 (half test set) | 4.84% | 2.69% |
| Pruned Llama2 (half test set) | 5.26% | 3.21% |

To strengthen our evaluation and provide additional validation of capability preservation, we conducted pairwise comparisons between the pruned and unpruned models. In this setup, we used the unpruned model's outputs as the reference point while still employing GPT-4 Turbo as the evaluator. The pruned model achieved a win rate of 51.49% in these comparisons, indicating essential parity with the unpruned model.

These results, particularly the near-50% win rate in pairwise comparisons, provide strong evidence that our pruning method successfully preserves model capabilities. While the standard evaluation shows a slight improvement, the key finding is the consistent performance between pruned and unpruned versions, as demonstrated across both evaluation approaches.

---

[7]https://github.com/tatsu-lab/alpaca_eval

## F.4 Evaluation on RULER Benchmark

To provide a more rigorous evaluation of our pruning method's impact on retrieval capabilities, we adopted the RULER benchmark Hsieh et al. (2024), which offers a comprehensive assessment framework beyond simple needle-in-haystack scenarios. We focused our evaluation on tasks specifically designed to test retrieval and copying abilities, including single and multi-key/value retrieval tasks (`niiah_single_1-3`, `niah_multikey_1-3`, `niah_multivalue`, and `niah_multiquery`).

We evaluated both the unpruned and pruned versions of LLaMA-3.1-Instruct Dubey et al. (2024) across different context lengths ranging from 4K to 128K tokens. Table 15 presents these results.

Table 15: Performance comparison between unpruned and pruned models on RULER benchmark tasks focused on retrieval capabilities.

| Model | Length | 4K | 8K | 16K | 32K | 64K | 128K | AVG |
|---|---|---|---|---|---|---|---|---|
| Unpruned | 128K | 99.9% | 99.9% | 99.8% | 99.6% | 98.7% | 92.6% | 98.4% |
| Pruned | 128K | 99.9% | 99.9% | 99.7% | 99.4% | 98.4% | 92.1% | 98.2% |

The results demonstrate that our pruning method maintains the model's retrieval capabilities across all context lengths. The pruned model achieves performance nearly identical to the unpruned version, with only a marginal difference of 0.2 percentage points in the average score (98.2% vs 98.4%). This minimal performance gap is particularly encouraging, as it suggests that our pruning approach effectively mitigates copy-bias while preserving the model's essential ability to accurately retrieve and reproduce information from the input context.

# G  BEYOND THE FEW-SHOT SETTINGS

To better understand the relationship between shot count and copying behavior, we conducted an expanded experiment examining performance beyond the few-shot setting. Figure 7 presents results averaged across all linguistic tasks over multiple seeds, with shot counts ranging from 1 to 64. Our findings demonstrate that the proposed pruning method yields consistent improvements across all shot counts, validating its effectiveness beyond the few-shot regime. However, we observe that the performance gap between pruned and unpruned models gradually narrows as the number of shots increases, with the most substantial improvements occurring in the 1-20 shot range for Bloom-560M. This pattern aligns with the intuition that as models receive more examples, they become better equipped to learn the underlying patterns rather than relying on copying behaviors. These results suggest that while copying bias naturally diminishes with increased examples, our pruning approach remains beneficial by promoting better pattern recognition over copying strategies.

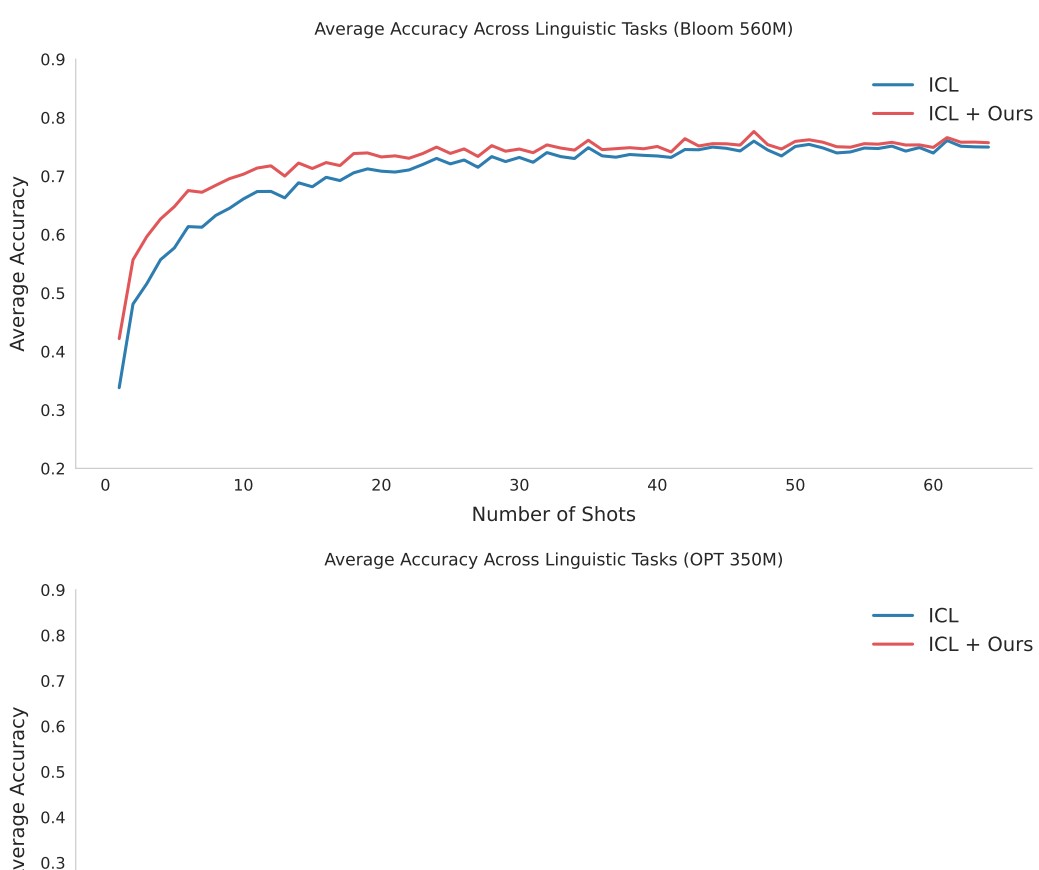

Figure 7: Many-Shots results for Bloom-560M and OPT-350M, results are averaged across all of the linguistic tasks

# H TASK VECTORS

We provide additional results for the task-vectors experiment, we include two additional models (GPT2-Small and Bloom1.1B) over the Algorithmic Next Letter task and Linguistics Antonyms. .

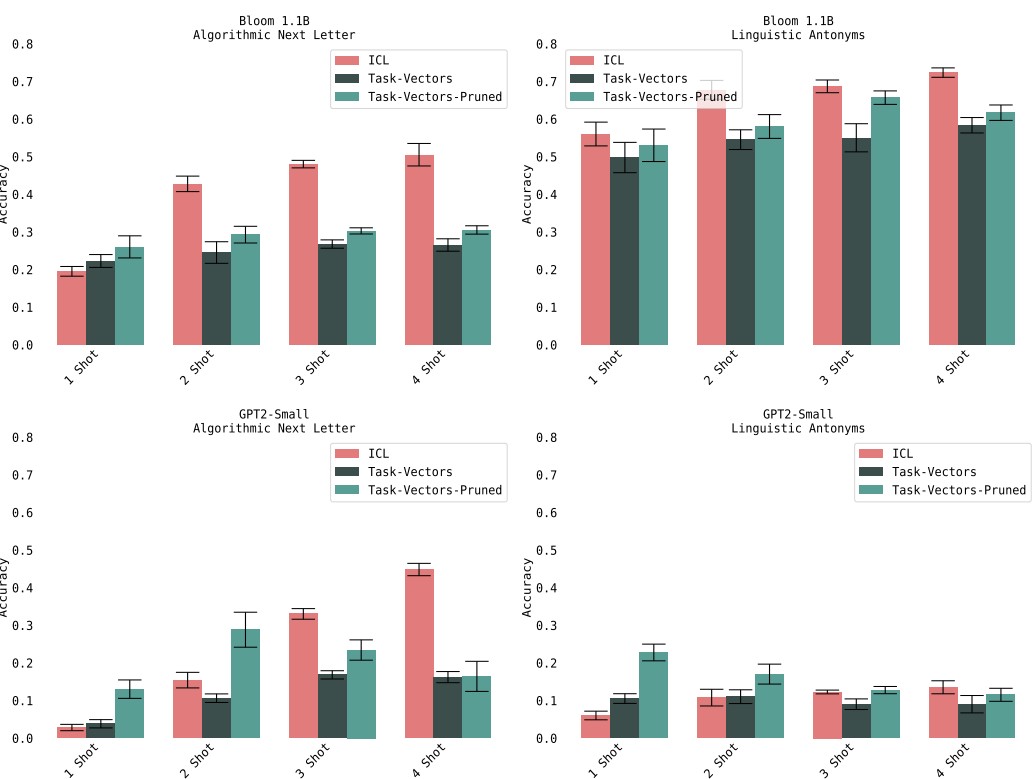

Figure 8: Additional quantitative results for the task-vectors experiment

# I ADDITIONAL PROXY TASK ANALYSIS

## I.1 PROXY DATASET ABLATION

To investigate the generality of our copying neuron detection approach, we conducted experiments using different proxy ICL tasks beyond vowel counting. We explored tasks from both linguistic (e.g., antonyms T11) and knowledge (e.g., person-language mapping T16) domains. Our results demonstrate that neurons identified using any of these alternative proxy tasks led to improvements across our evaluations.

Through these experiments, we identified three key criteria for an effective proxy task:

1. Clear, deterministic rules that require pattern recognition rather than memorization
2. Ability to elicit copying behavior from the model
3. Sufficiently constrained output space to reliably detect copying errors

While vowel counting meets these criteria particularly well and enables straightforward synthetic data generation, our results show it is not uniquely special - other structured tasks with similar properties can serve as effective proxies for copying neuron detection.

We evaluate the performance using different proxy tasks for neuron detection, with results averaged across several linguistic tasks (present simple to gerund, present to past simple, present to past perfect, and singular to plural). Tables 16 and 17 show the results using linguistic antonyms and knowledge person-language mapping as proxy tasks, respectively.

Table 16: Results using linguistic antonyms as proxy task. Each cell shows baseline | pruned accuracy.

| Model | 1 Shot | 2 Shot | 3 Shot | 4 Shot |
|---|---|---|---|---|
| GPT2 Small | 0.15\|**0.21** | 0.31\|**0.35** | 0.31\|**0.37** | 0.36\|**0.42** |
| GPT2 Medium | 0.24\|**0.30** | 0.39\|**0.47** | 0.43\|**0.52** | 0.50\|**0.58** |
| Bloom 560M | 0.33\|**0.37** | 0.40\|**0.48** | 0.48\|**0.53** | 0.51\|**0.58** |
| OPT 1.3B | 0.20\|**0.23** | 0.30\|**0.32** | 0.40\|**0.42** | 0.49\|**0.51** |

Table 17: Results using knowledge person-language as a proxy task. Each cell shows baseline | pruned accuracy.

| Model | 1 Shot | 2 Shot | 3 Shot | 4 Shot |
|---|---|---|---|---|
| GPT2 Small | 0.15\|**0.17** | 0.31\|**0.34** | 0.31\|**0.37** | 0.36\|**0.43** |
| GPT2 Medium | 0.24\|**0.27** | 0.39\|**0.43** | 0.43\|**0.48** | 0.50\|**0.55** |
| Bloom 560M | 0.33\|**0.38** | 0.40\|**0.49** | 0.48\|**0.56** | 0.51\|**0.61** |
| OPT 1.3B | 0.20\|**0.22** | 0.30\|**0.35** | 0.40\|**0.47** | 0.49\|**0.53** |

The consistent improvements across different proxy tasks and model architectures suggest that our method successfully identifies copying neurons regardless of the specific task used for detection, provided it meets our identified criteria.

## J  FURTHER ANALYSIS AND ABLATION STUDIES

### J.1  ROLE OF TASK DESCRIPTIONS

First, we investigate how explicit task descriptions interact with our copying neuron pruning method. Task descriptions provide explicit instructions about the required transformation (e.g., "Convert verbs from present to gerund form"). We compare three configurations: (1) standard ICL without task descriptions, (2) ICL with task descriptions, and (3) our pruning method, both with and without task descriptions.

Results show that while task descriptions improve baseline performance, our pruning method achieves substantially better results across all settings. Moreover, combining task descriptions with pruning (TD+Pruning) yields the best performance across all models and shot settings, suggesting these approaches are complementary in addressing different aspects of the copying bias problem.

Table 18 shows results averaged across linguistic tasks (present simple to gerund, present simple to past simple, present simple to past perfect, singular to plural) over 5 different seeds.

Table 18: Impact of Task Descriptions (TD) and Pruning across different model sizes

| Model | 1 Shot | 2 Shot | 3 Shot | 4 Shot |
|---|---|---|---|---|
| GPT2-Small+ICL | 0.16 | 0.31 | 0.32 | 0.37 |
| GPT2-Small+ICL+TD | 0.19 | 0.35 | 0.37 | 0.41 |
| GPT2-Small+ICL+Pruning | 0.25 | 0.44 | 0.44 | 0.47 |
| GPT2-Small+ICL+TD+Pruning | **0.27** | **0.45** | **0.46** | **0.50** |
| Bloom-560M+ICL | 0.35 | 0.42 | 0.48 | 0.50 |
| Bloom-560M+ICL+TD | 0.37 | 0.46 | 0.43 | 0.55 |
| Bloom-560M+ICL+Pruning | 0.42 | **0.53** | 0.59 | 0.59 |
| Bloom-560M+ICL+TD+Pruning | **0.44** | **0.53** | **0.61** | **0.60** |
| Llama3-8B+ICL | 0.68 | 0.85 | 0.88 | 0.89 |
| Llama3-8B+ICL+TD | 0.70 | 0.88 | 0.89 | **0.90** |
| Llama3-8B+ICL+Pruning | **0.73** | 0.89 | **0.90** | **0.90** |
| Llama3-8B+ICL+TD+Pruning | **0.73** | **0.90** | **0.90** | **0.90** |

### J.2  COMPARISON WITH ALTERNATIVE TRAINING METHODS

To better understand the effectiveness of our pruning approach, we compare it with traditional fine-tuning on the vowel mapping task. We conduct a comprehensive hyperparameter sweep for the fine-tuning baseline, exploring learning rates [1e-5, 1e-4, 1e-3], batch sizes [4, 8, 16], and epochs [1, 3, 5], selecting the best configuration using the validation set.

Table 19 presents results averaged across all linguistic tasks for 5 different seeds, comparing baseline performance with both fine-tuning and our pruning approach.

### J.3  BALANCED DEMONSTRATION ANALYSIS

To investigate whether our method addresses fundamental copying mechanisms rather than dataset biases, we evaluate performance on SST2 using balanced demonstrations with equal representation of positive and negative examples. Results in Table 20 demonstrate that the improvements from our pruning method persist even in this controlled setting. These experiments demonstrate several key points about our pruning method: (1) it provides complementary benefits when combined with task descriptions, (2) it outperforms traditional fine-tuning approaches while requiring no gradient updates, and (3) it addresses fundamental copying mechanisms rather than surface-level dataset biases. The consistent improvements across different settings and model sizes suggest that our approach successfully targets a core aspect of model behavior.

Table 19: Comparison of pruning with traditional fine-tuning approaches

| Model | 1 Shot | 2 Shot | 3 Shot | 4 Shot |
|---|---|---|---|---|
| Bloom-560M Baseline | 0.35 | 0.42 | 0.48 | 0.50 |
| Bloom-560M Finetuned | 0.34 | 0.41 | 0.45 | 0.50 |
| Bloom-560M Ours | **0.42** | **0.53** | **0.59** | **0.59** |
| GPT2-Small Baseline | 0.16 | 0.31 | 0.32 | 0.37 |
| GPT2-Small Finetuned | 0.14 | 0.31 | 0.32 | 0.35 |
| GPT2-Small Ours | **0.25** | **0.44** | **0.44** | **0.47** |
| OPT-1.3B Baseline | 0.21 | 0.30 | 0.42 | 0.50 |
| OPT-1.3B Finetuned | 0.20 | 0.30 | 0.42 | 0.50 |
| OPT-1.3B Ours | **0.24** | **0.42** | **0.52** | **0.59** |

Table 20: Performance on SST2 with balanced demonstrations

| Model | 2 Shot | 4 Shot | 6 Shot |
|---|---|---|---|
| Llama2-7B ICL | 0.873 | 0.920 | 0.923 |
| Llama2-7B ICL Balanced | 0.881 | 0.920 | 0.924 |
| Llama2-7B ICL Ours | **0.898** | **0.922** | **0.926** |

### J.4 IMPACT OF MODEL PRE-TRAINING OBJECTIVES

To understand how different pre-training objectives affect copying bias, we analyze the effectiveness of our pruning method on both base and instruction-tuned variants of the same model architecture. We compare OPT-1.3B base model with its instruction-tuned counterpart[8].

Table 21 shows results averaged across linguistic tasks (present simple to gerund, present simple to past simple, present simple to past perfect, singular to plural, and antonyms) over 5 different seeds. For each model and shot setting, we report both baseline and pruned performance.

Table 21: Performance comparison between base and instruction-tuned models

| Model | 1 Shot | 2 Shot | 3 Shot | 4 Shot |
|---|---|---|---|---|
| OPT-1.3B Base | 0.28\|**0.32** | 0.38\|**0.48** | 0.49\|**0.57** | 0.55\|**0.62** |
| OPT-1.3B Instruct | 0.27\|**0.32** | 0.39\|**0.46** | 0.51\|**0.57** | 0.55\|**0.64** |

Results demonstrate that our pruning method yields consistent improvements regardless of the model's pre-training objective. Both base and instruction-tuned variants show similar relative gains across different shot settings, suggesting that copying bias and the effectiveness of our mitigation strategy are inherent to the model architecture rather than being specific to particular training regimes.

---

[8]We use the publicly available OPT-IML-1.3B model from https://huggingface.co/facebook/opt-iml-1.3b

