# OpenReview forum: "Mitigating Copy Bias in In-Context Learning through Neuron Pruning"
_ICLR.cc/2025/Conference — Submitted to ICLR 2025_

### Official Review · Reviewer_4ESn · 2024-10-28

**Soundness:** 2
**Presentation:** 2
**Contribution:** 1
**Rating:** 5
**Confidence:** 4

**Summary:**

This paper shows that large language models (LLMs) can exhibit a copying bias, often simply copying their answers from the in-context examples. The authors identified task-agnostic neurons responsible for this copying behavior using integrated gradients [1] on a vowel-counting proxy task (Secs. 3.2 & 3.3). They mitigate their influence by pruning them (i.e., simply setting them to 0; Eq. 8). Their neuron pruning method improves in-context learning (ICL) performance across various tasks, including the ICL benchmark proposed by Hendel et al [2], two sentiment analysis tasks and an object counting task, and various models, including GPT-2, Llama variants or Mamba. They also find that pruning these neurons improves the effectiveness of task vectors [2].

---

[1] Sundararajan, M., Taly, A., & Yan, Q. (2017). Axiomatic attribution for deep networks. In ICML.

[2] Hendel, R., Geva, M., & Globerson, A. (2023). In-context learning creates task vectors. In EMNLP.

**Strengths:**

* **S1**: Previous work has focused on improving ICL primarily through black-box approaches (e.g., prompt tuning), whereas this work adopts a mechanistic perspective. This shift is valuable, as it has the potential to deepen our understanding of the internal workings of ICL in these models.

* **S2**: The pruning method effectively improves ICL performance across various architectures and tasks. Additionally, it enhances the quality of task vectors, also leading to improvements in ICL performance.

**Weaknesses:**

* **W1**: The overall technical contributions (Sec. 3) are modest. The work applies a well-established attribution method (integrated gradients) to identify neurons responsible for a specific behavior (copying) and then prunes these neurons to achieve the intended outcome (no copying, though that’s unclear to me, see **W2**). Unfortunately, this is highly similar to prior work such as the one by Ghorbani et al [1]. For example, they used Shapley values (another attribution method) to identify neurons that led to gender-specific biases and pruned them to achieve gender fairness. Thus, while the application to copying in ICL is novel, the methodological contributions remain limited (i.e., the use of prediction shift ΔL instead of max probability for integrated gradients, l. 227-230).

* **W2**: It is unclear to me whether the performance improvement results from the reduction of copying neurons or from the pruning of neurons exhibiting different behavior. For example, why would performance improve on *copying* tasks such as “list first” or “list max” in the ICL benchmark of Hendel et al [2], which require copying from the in-context examples? Tabs. 3-6 consistently also show improvements on these *copying* tasks. If the identified neurons indeed suppress the copying behavior, we would expect to see drops in performance on such tasks. This seems to contradict one of the authors’ main claims that they “identify specific neurons responsible for triggering the copying behavior (copying neurons)” (l. 71-73).

* **W3**: It is unclear whether the copying bias is merely an artifact of the ill-posed nature of few-shot ICL problems, and whether it remains relevant for many-shot ICL problems. Does this copying bias persist beyond the few-shot setting? (This point is also raised in the questions below.)

* **W4**: The writing could be improved. There are several typos throughout, and clarity and reproducibility would benefit from including all necessary details, such as those regarding proxy dataset generation.

* **W5**: Code is not provided, making reproducibility challenging.

---

[1] Ghorbani, A., & Zou, J. Y. (2020). Neuron shapley: Discovering the responsible neurons. NeurIPS.

[2] Hendel, R., Geva, M., & Globerson, A. (2023). In-context learning creates task vectors. In EMNLP.

**Questions:**

* **Q1**: Is the copying bias merely an artifact of having a limited number of examples (e.g., <= 4)? For instance, what happens in a 64-shot setting? Could this bias indicate that the copying issue arises from an ill-posed in-context learning (ICL) problem rather than a general model flaw?

* **Q2**: Could the authors provide additional details about the proxy dataset? For example, how many shots are used, and could a few more examples be included for completeness?

* **Q3**: How does pruning of these neurons affect the overall model performance beyond ICL tasks? For example, how do the pruned LLMs perform on more general benchmarks such as [Live-Bench](https://livebench.ai/) or other reasoning benchmarks (e.g., MMLU or GSM8K)? Or, what’s the performance on the indirect object identification (IOI) task? The authors suggest that their method “could be broadly applicable, potentially becoming a standard post-processing step in LLM deployment pipelines” (l. 497-498). However, it remains unclear if this would be a sensible approach for LLMs deployed “in the wild.”

* **Q4**: Are the minimum and maximum values of the normalizing functions calculated across the dataset alone, or are they based on both the dataset and all neurons?

* **Q5**: What’s the significance of the meta-optimization (l. 319-369)? Why not greedily remove the neurons based on the “copying score” from Eq. 7 (this may be related to the question in **Q4**)?

* **Q6**: What’s the relation of the real-word experiments in Sec. 4.2 (sentiment analysis and object counting) to the copying behavior that’s the main topic of the present work?

---

> ### Author Response · Authors · 2024-11-19
> **Response 1/4**
>
> We thank the reviewer for their comments.
>
> > Q1: Is the copying bias merely an artifact of having a limited number of examples (e.g., <= 4)? For instance, what happens in a 64-shot setting? Could this bias indicate that the copying issue arises from an ill-posed in-context learning (ICL) problem rather than a general model flaw?
>
> Our additional experiments with up to 64 shots (reported in Appendix G) directly address this concern. While copying bias is indeed more pronounced in few-shot settings, our results show it persists even with increased examples, albeit to a lesser degree. The gradual narrowing of the performance gap between pruned and unpruned models as shot count increases suggests that copying bias is not merely an artifact of limited examples but rather a fundamental behavior that models naturally rely on. This behavior diminishes as models receive more examples and become better equipped to learn underlying patterns, yet our pruning method continues to yield consistent improvements even in many-shot settings. This indicates that while the problem is more acute in few-shot scenarios, copying bias represents a general model tendency that our method effectively mitigates across different shot regimes.
>
> >Q2: Could the authors provide additional details about the proxy dataset? For example, how many shots are used, and could a few more examples be included for completeness?
>
> In our implementation, we have used n=2 examples per prompt which allows us to study copying behavior. Our synthetic dataset construction addresses the concerns about label repetition: we ensure that each prompt's examples (Sp) contain distinct vowel counts, and the target word's vowel count (ŷ) is always different from those in Sp. We sample words with vowel counts ranging from 1-8, providing sufficient diversity while avoiding repetition within prompts. These constraints are applied during dataset generation. Examples from our proxy dataset:
>
> * "bed → 1"
> * "school → 2"
> * "octopus → 3"
> * "penguin → “3"
> * "zebra → 2"
>
> We generate 400 prompts, we use 300 for the detection process and 100 for the meta-optimization ( a validation set to determine optimal block and pruning percentage). We have updated the revised to include these details. We also refer the reviewer to the newly added Appendix I, where we explore alternative proxy tasks, demonstrating that the specific choice of the proxy task may not be as critical.
>
> >Q3: How does pruning of these neurons affect the overall model performance beyond ICL tasks? For example, how do the pruned LLMs perform on more general benchmarks such as Live-Bench or other reasoning benchmarks (e.g., MMLU or GSM8K)? Or, what’s the performance on the indirect object identification (IOI) task? The authors suggest that their method “could be broadly applicable, potentially becoming a standard post-processing step in LLM deployment pipelines” (l. 497-498). However, it remains unclear if this would be a sensible approach for LLMs deployed “in the wild.”

---

> ### Author Response · Authors · 2024-11-19
> **Response 2/4**
>
> >Q3: How does pruning of these neurons affect the overall model performance beyond ICL tasks? For example, how do the pruned LLMs perform on more general benchmarks such as Live-Bench or other reasoning benchmarks (e.g., MMLU or GSM8K)? Or, what’s the performance on the indirect object identification (IOI) task? The authors suggest that their method “could be broadly applicable, potentially becoming a standard post-processing step in LLM deployment pipelines” (l. 497-498). However, it remains unclear if this would be a sensible approach for LLMs deployed “in the wild.”
>
> To assess potential tradeoffs introduced by our pruning method, we evaluate the pruned models on perplexity (WikiText) and MMLU benchmark. As shown in the following two tables, our method has minimal impact on these capabilities, while MMLU performance remains stable or even improves in some cases, indicating that our targeted neuron pruning successfully mitigates copying behavior while preserving general capabilities. We have updated the draft to include these results with additional models in Appendix F.
>
> Perplexity on WikiText test (lower is better) :
> |    Model   | Without pruning | With Pruning |
> |:----------:|:---------------:|:------------:|
> |  GPT-Small |      **25.188**     |    25.548    |
> | GPT-Medium |      **18.473**     |    18.590    |
> | Bloom 560M |     **21.328**     |    21.397    |
> | Bloom 1.1B |      **16.836**     |    16.883    |
> |  OPT 125m  |      **26.119**     |    26.432    |
> |  OPT 350m  |      **20.888**     |    20.940    |
> |  OPT 1.3B  |      **13.889**    |    13.904    |
> |  OPT 2.7B  |      **11.757**     |    11.758    |
> |  Llama2 7b |      **6.5420**    |    6.5490    |
> | Llama3 78b |      **5.1840**     |    5.8190    |
>
> MMLU (higher is better):
>
> |       Model      | humanities | social sciences |  stem  |  other |
> |:----------------:|:----------:|:---------------:|:------:|:------:|
> |    Gpt2 medium   |   0.2290   |      **0.2427**     | **0.2350** | 0.2128 |
> | Gpt2 medium+ours |   **0.2427**   |      0.2379     | 0.2294 | **0.2144** |
> |    Bloom 1.1b    |   0.2482   |      **0.2626**     | 0.2252 | 0.2347 |
> |  Bloom 1.1b+ours |   **0.2502**   |      0.2620     | **0.2268** | **0.2372** |
> |     Opt 2.7B     |   **0.2670**   |      0.2398     | **0.2470** | **0.2688** |
> |  Opt 2.7B + ours |   0.2648   |      **0.2427**     | 0.2457 | 0.2677 |
> |    Llama 2 7b    |   **0.4329**   |      0.5484     | 0.5297 | **0.3606** |
> |  Llama 2 7b+ours |   0.4327   |      **0.5491**     | 0.5297 | 0.3603 |
> |    Llama 3 8b    |   **0.5501**   |      **0.7078**     | 0.7322 | 0.5373 |
> |  Llama 3 8b+ours |   0.5486   |      0.7048     | **0.7329** | **0.5375** |
>
> >Q4: Are the minimum and maximum values of the normalizing functions calculated across the dataset alone, or are they based on both the dataset and all neurons?
>
> For each neuron, we first compute its attribution scores using integrated gradients across all prompts in our detection synthetic dataset. We then normalize these attribution scores for each neuron independently by applying min-max normalization across the scores of the different prompts (using the minimum and maximum values of that neuron's scores across different prompts). Finally, we average these normalized scores to obtain the final relevance score for the neuron. This ensures we get comparable scores across different samples before aggregation.
> We have updated Equation 7 in the revised version to make it more clear.
>
> >Q5: What’s the significance of the meta-optimization (l. 319-369)? Why not greedily remove the neurons based on the “copying score” from Eq. 7 (this may be related to the question in Q4)?
>
> The meta-optimization process (using a validation set to determine optimal block and pruning percentage) is crucial rather than simply removing neurons based on the copying score for several reasons. First, while the copying score identifies neurons that contribute to copying behavior, it doesn't directly tell us the optimal proportion to remove - pruning too many neurons could impair the model's general capabilities, while pruning too few might not sufficiently mitigate the copying bias. Second, different model architectures may have different optimal pruning points due to their varying architectures and redundancies. The meta-optimization helps us find this balance automatically.

---

> > ### Author Response · Authors · 2024-11-19
> > **Response 3/4**
> >
> > > Q6: What’s the relation of the real-word experiments in Sec. 4.2 (sentiment analysis and object counting) to the copying behavior that’s the main topic of the present work?
> >
> > 1. Demonstration of Real-World Copying Issues:
> >
> > Sentiment analysis and object counting are tasks where copying from examples can be particularly problematic.
> >
> > * In sentiment analysis, models may copy ratings from examples with similar words/phrases rather than analyzing the actual sentiment.
> >
> > * In object counting, models may copy counts from examples with similar object mentions rather than actually counting
> >
> > These provide concrete examples of how copying bias manifests in practical applications
> >
> > 2. Validation Beyond Synthetic Tasks:
> >
> > While our method is developed using synthetic tasks (vowel counting), these experiments show it generalizes to:
> >
> > * Real-world data distributions.
> > * More complex reasoning tasks.
> > * Practical application scenarios.
> >
> > Additionally, these tasks have established baselines (WICL, APE) focusing on improving ICL. Showing our method outperforms these baselines suggests: (1) Copying bias is a significant factor in ICL performance, (2) Addressing copying specifically is more effective than general ICL improvements, (3) Our method successfully targets a fundamental ICL limitation.
> >
> > >W1: The overall technical contributions (Sec. 3) are modest. The work applies a well-established attribution method (integrated gradients) to identify neurons responsible for a specific behavior (copying) and then prunes these neurons to achieve the intended outcome (no copying, though that’s unclear to me, see W2). Unfortunately, this is highly similar to prior work such as the one by Ghorbani et al [1]. For example, they used Shapley values (another attribution method) to identify neurons that led to gender-specific biases and pruned them to achieve gender fairness. Thus, while the application to copying in ICL is novel, the methodological contributions remain limited (i.e., the use of prediction shift ΔL instead of max probability for integrated gradients, l. 227-230).
> >
> > Thank you for pointing us to [1]. There are numerous differences between our work and theirs beyond the use of different explainability methods. The relevant experiment in [1] removes filters from a CNN that is trained on one dataset to classify gender, when it is applied to another, more balanced dataset. The per-filter scores are computed on the 2nd dataset, as far as we can tell (a bit unclear), and those neurons that are causal for the most mistakes are removed. We work in the domain of NLP, transformers and not CNNS, work with neurons and not filters, and make sure to identify the neurons contributing to the general copying issue (as opposed to a task-specific failure) by using a made-up synthetic task (and not the same dataset or even the same task). Based on these differences, it does not seem that [1] takes away much of our novelty.

---

> ### Author Response · Authors · 2024-11-19
> **Response 4/4**
>
> > W2: It is unclear to me whether the performance improvement results from the reduction of copying neurons or from the pruning of neurons exhibiting different behavior. For example, why would performance improve on copying tasks such as “list first” or “list max” in the ICL benchmark of Hendel et al [2], which require copying from the in-context examples? Tabs. 3-6 consistently also show improvements on these copying tasks. If the identified neurons indeed suppress the copying behavior, we would expect to see drops in performance on such tasks. This seems to contradict one of the authors’ main claims that they “identify specific neurons responsible for triggering the copying behavior (copying neurons)” (l. 71-73).
>
> 1. Clarifying "Copying Errors" vs. "Valid Pattern Matching":
>
> Our work targets inappropriate copying where the model reproduces an promt example answer instead of applying the learned pattern This is different from tasks like "list first" or "list max" which require understanding and applying a consistent pattern that may involve selecting/copying elements from the example itself according to a rule. When we say "copying neurons," we specifically mean neurons that trigger direct reproduction of example outputs (and not example content)  without pattern recognition.
>
> 2. Evidence for This Distinction:
> For tasks like "list first," the model must: Understand the positional pattern (select first element). Apply this pattern to new inputs. "Copy" the correct element based on the learned rule.
> In contrast, copying errors occur when the model: Fails to learn the underlying pattern. Instead directly copies an example output regardless of whether it follows the task pattern
>
> The neurons we identify through ΔL specifically target cases where probability mass shifts toward example outputs that don't match the ground truth. This focuses on inappropriate copying while preserving the model's ability to learn and apply patterns that may involve selective copying. The improvement on "copying tasks" suggests our method is successfully distinguishing between: Valid pattern-based selection/copying, Invalid memorization-based copying.
>
> If we were indiscriminately suppressing all copying behavior, we would indeed expect degradation on tasks requiring valid copying. The consistent improvements across both pattern-matching tasks and other tasks suggest we're successfully targeting problematic copying while preserving valid pattern learning.
>
> >W3: It is unclear whether the copying bias is merely an artifact of the ill-posed nature of few-shot ICL problems, and whether it remains relevant for many-shot ICL problems. Does this copying bias persist beyond the few-shot setting? (This point is also raised in the questions below.)
>
> Our additional experiments with up to 64 shots (see Appendix G) reveal that while copying bias is most prominent in few-shot settings, it doesn't completely disappear with more examples. As the number of shots increases, we observe a smaller but still meaningful gap between pruned and unpruned models. This pattern suggests that copying isn't just a fallback strategy when examples are scarce - it's an inherent model behavior that gradually gives way to pattern learning as more examples become available. The persistent improvements from our pruning method, even with many shots, confirm we're addressing a fundamental aspect of how these models process in-context examples.
>
> >W4: The writing could be improved. There are several typos throughout, and clarity and reproducibility would benefit from including all necessary details, such as those regarding proxy dataset generation.
> Fixed and updated at the relevant places in the paper draft.
>
> W5: Code is not provided, making reproducibility challenging.
> Code will be made publicly accessible

---

> > ### Author Response · Authors · 2024-11-23
> >
> > Dear Reviewer 4ESn
> >
> > As we approach the end of this discussion period, we want to thank you for your thoughtful feedback, which has significantly improved our paper. We believe we’ve addressed most of the concerns raised by you and the other reviewers. In particular, we have incorporated:
> >
> > - Additional implementation details.
> > - Evaluations across a broader range of tasks (e.g., MMLU, AlpacaEval2) and retrieval challenges like Needle In A Haystack
> > - Ablation studies exploring the effectiveness of our method beyond the few shot settings.
> >
> > These experiments demonstrate that our approach consistently mitigates copy bias without compromising performance on tasks where copying is beneficial (e.g., certain retrieval tasks).
> >
> > We would appreciate it if you could let us know whether our responses properly address your concerns.

---

> > > ### Comment · Reviewer_4ESn · 2024-11-24
> > >
> > > I would like to thank the authors for their responses to my initial and other reviewer’s reviews and ongoing discussions. I want to particularly thank them for including ICL experiments with more shots, general language modeling assessments, and added clarifications in the revision.
> > >
> > > I’ll raise my score from 3 to 5. However, I’ll remain on the reject side because I still think that *methodological* novelty is limited (while I agree with the authors that the application is novel, I find the methodological contributions to be small/incremental) and the paper would be strengthened by conducting a more thorough (mechanistic) analysis of “copying neurons” (to better understand their effect on the models’ behavior).

---

> > > > ### Author Response · Authors · 2024-11-28
> > > >
> > > > Thank you for raising our score and for the detailed discussion of our work. However, we respectfully disagree with your point regarding the lack of insights into why the improvements occur.
> > > >
> > > > This comment echoes your earlier observation in W2: "If the identified neurons indeed suppress the copying behavior, we would expect to see drops in performance on such tasks."
> > > >
> > > > Our findings demonstrate that the pruned model produces more accurate task vectors (see figure 5, page 9), effectively driving the model to focus on task detection rather than using copying as a shortcut. This aligns with our results, which show that pruning mitigates copy bias—preventing copying when it is detrimental—while preserving it when it is genuinely beneficial (e.g., as evidenced by the Retrieval / Needle-in-a-Haystack experiments). These additional experiments, which we have now incorporated, further support this conclusion. We believe this connection between neuron pruning and improved task detection/task vectors is both meaningful and insightful.
> > > >
> > > > That said, we are uncertain whether a further  mechanistic interpretation of individual neurons is feasible or would provide additional insights. The pruned neurons do not appear interpretable in isolation. This does contrast with CNNs, where filters are often easier to understand and visualize (e.g., pruned filters in Ghorbani and Zou, NeurIPS 2020) but, in itself, may be not so surprising in LLM/Transformers, at neuron level. While individual interpretable neurons for specific models might have been possible, we believe this approach risks being misleading and would not yield generalizable insights. Note that this is not different for most work on subnetwork pruning in LLMs, individual neurons and weights do not typically have an interpretable role.
> > > > Regarding the novelty of our technical contribution, we acknowledge that our approach is simple – a modification of Integrated Gradients with logit-difference for pruning – and related to the work of Ghorbani and Zou (NeurIPS 2020). However, their method is computationally infeasible in our setting.
> > > >
> > > > We argue that a focused study of this method's effectiveness within the context of ICL is valuable in itself. Notably, we find it intriguing and somewhat surprising that this simple method identifies highly functionally localized behavior. Furthermore, its performance remains unaffected across other tasks, including those outside ICL (see our discussion with Reviewer 9Q7b). Together with comparisons to standard fine-tuning, this suggests our method could be a strong candidate for broader applications – a key insight for practitioners. Its simplicity, we hope, will further encourage adoption.

---

### Official Review · Reviewer_ypEw · 2024-11-03

**Soundness:** 3
**Presentation:** 3
**Contribution:** 3
**Rating:** 8
**Confidence:** 4

**Summary:**

This paper addresses the issue of copying bias in LLMs during in-context learning (ICL). To mitigate copying-biase, the authors identify and prune specific "copying neurons". They use Integrated Gradients (IG) on a synthetic vowel-counting task to locate these neurons and then prune them. The approach is shown to generalize across multiple model architectures (e.g., GPT-2, OsPT, BLOOM) and improves the quality of task vectors—representations that encapsulate task-specific information—indicating better task recognition. This method provides an effective way to enhance ICL performance without task-specific fine-tuning, offering a broadly applicable strategy for deploying more reliable LLMs.

**Strengths:**

1. This paper demonstrates that mechanistic interpretability methods like Integrated Gradients (IG), typically used for understanding model behaviors, can have practical implications such as improving in-context learning (ICL). By identifying and pruning neurons responsible for copying behavior, the paper presents an innovative approach to enhancing ICL performance.

2. The authors conduct experiments across a wide range of tasks and model architectures, demonstrating consistent performance improvements. This thorough evaluation reinforces the robustness and generalizability of the proposed method.

3.  I appreciate that authors can first justify the prevalence of copying errors to justify why such a bias is worth targeting across different models and tasks. This rigorous approach highlights the significance of addressing this bias, underscoring the method’s relevance and value in improving model generalization.

**Weaknesses:**

1. **Similarity of Tasks and Limited Generalizability**: The 18 tasks studied in this paper share notable similarities with vowel counting, meaning that the copying neurons identified through Integrated Gradients primarily apply to a set of very related and rather synthetic tasks. While the authors also attempt to show improvements on non-synthetic tasks, I am not sure how to interpret "copying error" on datasets like SST2 and SST5, which are multiple-choice tasks with a limited set of choices (e.g., 2 or 5 options)? In such cases, does the model exhibit a genuine copying bias or if the improvements are driven by other factors? Would the copying bias be easily mitigated by using unbiased ICL samples (for example, one sample for each answer choice A/B/C/D).

2. **Limited Exploration of Trade-offs**: The paper does not extensively address the potential trade-offs introduced by pruning neurons. Authors only claim that the pruned model never has a drop in performance but what about other side effects? For instance, pruning could theoretically affect the model's performance on tasks where copying might be beneficial (e.g., fact-based retrieval tasks). A more detailed analysis of cases where pruning could have a negative impact would be beneficial.

3. **Minor Reproducibility Concerns**: The paper lacks details on certain experimental settings, such as the dataset specifications (e.g., training/validation sizes, number of shots) for the vowel-counting data, as well as the hyperparameter settings used for pruning. Including these details would enhance reproducibility and provide greater clarity on the experimental setup.

**Questions:**

1. In Figure 2, it’s clear that the copying error rate has dropped consistently across different models and tasks, contributing to improved accuracy. However, I’m surprised that copying errors still dominate the error rate in the unpruned model. Do the authors have any insight into why this phenomenon occurs? Additionally, how does the copying error rate compare for the vowel-counting task specifically? I’m curious about the extent to which copying errors can be reduced on the in-distribution task (vowel counting) versus out-of-distribution tasks (the other ICL tasks).”

2. Why are all the tasks restricted to 4 shots? My intuition is that with more few shot examples, the model is more likely to pick up the actual pattern and less likely to copy, therefore, the method will become less effective. This analysis is not  to weaken the method — as authors have pointed out that some task might have limited ICL examples as a restriction, and the proposed method is a great way to mitigate the bias. I am trying to get a better understanding on the limitation of the proposed method.

3. For the BBH object counting task, my understanding is that these are complex reasoning tasks that benefit from Chain-of-Thought (CoT) reasoning, which ICL examples could potentially demonstrate. Can the authors clarify if CoT prompting is employed in either the baseline models or the copy-error-pruned method?

---

> ### Author Response · Authors · 2024-11-19
> **Response 1/3**
>
> Thank you for your effort and for taking the time to help us improve our paper.
>
> > In Figure 2, it’s clear that the copying error rate has dropped consistently across different models and tasks, contributing to improved accuracy. However, I’m surprised that copying errors still dominate the error rate in the unpruned model. Do the authors have any insight into why this phenomenon occurs? Additionally, how does the copying error rate compare for the vowel-counting task specifically? I’m curious about the extent to which copying errors can be reduced on the in-distribution task (vowel counting) versus out-of-distribution tasks (the other ICL tasks).”
>
> We believe copying errors dominate because it's a "safer" failure mode for the model - repeating a previously seen answer rather than generating a potentially incorrect novel response. This aligns with the training objective of minimizing unlikely outputs. For the vowel-counting task, copying errors occur at a similar rate to other tasks, suggesting that copying behavior is a general phenomenon rather than task-specific. This consistency between in-distribution and out-of-distribution tasks supports our hypothesis that we're identifying fundamental copying mechanisms in the model architecture rather than task-specific patterns.
>
> In the following table, we report (Copying Error Rate / Total Error Rate) over the vowel counting for a single run. (we have updated the revised version to include these results in Appendix C)
> |    Model    | 1 Shot | 2 Shot | 3 Shot | 4 Shot |
> |:-----------:|:------:|:------:|:------:|:------:|
> |  GPT2-Small |  0.936 |  0.566 |  0.669 |  0.733 |
> | GPT2-Medium | 0.853 |  0.430 |  0.711 |  0.812 |
> |  Bloom 560M |  0.751 |  0.390 |  0.580 |  0.686 |
> |  Bloom 1.1B |  0.987 |  0.680 |  0.777 |  0.906 |
> |   OPT 125M  |  0.802 |  0.657 |  0.689 |  0.680 |
> |   OPT 350M  |  0.227 |  0.775 |  0.812 |  0.872 |
> |   OPT 1.3B  |  0.951 |  0.378 |  0.682 |  0.80  |
> |   OPT 2.7B  |  0.942 |  0.439 |  0.621 |  0.712 |
> |  Llama3 8B  |  0.835 |  0.500 |  0.691 |  0.792 |
>
> >Why are all the tasks restricted to 4 shots? My intuition is that with more few shot examples, the model is more likely to pick up the actual pattern and less likely to copy, therefore, the method will become less effective. This analysis is not to weaken the method — as authors have pointed out that some task might have limited ICL examples as a restriction, and the proposed method is a great way to mitigate the bias. I am trying to get a better understanding on the limitation of the proposed method.
>
> To address this, we conducted additional experiments beyond 4 shots, testing performance with up to 64 shots (updated and reported in Appendix G). The results confirm the reviewer's intuition: as models receive more examples, they indeed become better at learning the underlying patterns rather than relying on copying behaviors, leading to a gradually narrowing performance gap between pruned and unpruned models. However, our pruning method continues to yield consistent improvements even in many-shot settings, albeit with diminishing returns. This pattern suggests that while our method is particularly powerful in few-shot scenarios (where copying bias is most problematic and many real-world applications are constrained), it remains beneficial across different shot regimes by promoting better pattern recognition over copying strategies. We believe this analysis provides a clearer understanding of our method's scope and limitations while reinforcing its practical value in scenarios where example availability is constrained.
>
> >For the BBH object counting task, my understanding is that these are complex reasoning tasks that benefit from Chain-of-Thought (CoT) reasoning, which ICL examples could potentially demonstrate. Can the authors clarify if CoT prompting is employed in either the baseline models or the copy-error-pruned method?
>
> In our experiments, we used simple input-output format for all methods to isolate the impact of our pruning approach. While CoT prompting is indeed powerful for complex reasoning tasks like BBH, we observed that copying errors still occur even with CoT - for example, in our vowel counting task, models sometimes copy incorrect vowel counts from demonstration examples even when chain-of-thought reasoning is provided. In additional experiments with Llama3, we found that while Chain-of-Thought prompting improved standard ICL performance by 1.5% in the 1 Shot settings for BBH, our pruning method achieved a more substantial 6% improvement. Future work could explore how our pruning method complements CoT and other advanced prompting techniques.

---

> ### Author Response · Authors · 2024-11-19
> **Response 2/3**
>
> >Similarity of Tasks and Limited Generalizability: The 18 tasks studied in this paper share notable similarities with vowel counting, meaning that the copying neurons identified through Integrated Gradients primarily apply to a set of very related and rather synthetic tasks. While the authors also attempt to show improvements on non-synthetic tasks, I am not sure how to interpret "copying error" on datasets like SST2 and SST5, which are multiple-choice tasks with a limited set of choices (e.g., 2 or 5 options)? In such cases, does the model exhibit a genuine copying bias or if the improvements are driven by other factors? Would the copying bias be easily mitigated by using unbiased ICL samples (for example, one sample for each answer choice A/B/C/D).
>
> Task Diversity Beyond Syntactic Similarity:
>
> While some tasks share surface-level similarities with vowel counting, they require fundamentally different reasoning: (1) Vowel counting: Character-level pattern recognition, (2) Linguistic tasks: Understanding morphological rules (e.g., singular-plural) and (3) Knowledge tasks: Retrieving factual relationships (e.g., country-capital)
>
> The success across this diverse range suggests we're capturing a general copying mechanism rather than task-specific patterns.
> SST and Multiple-Choice Tasks: We can observe copying bias in sentiment analysis even with limited choices:
> Models may copy ratings from examples with similar phrases rather than analyzing sentiment, Example: Given a prompt with:
>
> "The movie was okay but boring": Negative
>
> "This film bored me to tears": Negative
>
> "The ending was boring but had good moments": ???
>
> A copying-biased model may predict "Negative" based on the word "boring" appearing in previous examples, rather than analyzing the nuanced sentiment
>
> Our pruning improves the model's ability to consider the full context rather than copying based on lexical overlap.
> To verify this, we conducted additional experiments on SST2 using balanced ICL demonstrations (equal number of positive and negative examples), results are averaged across 5 different seeds:
>
> | Model                  | 2 Shots | 4 Shots | 6 Shots |
> |------------------------|---------|---------|---------|
> | Llama2 7B ICL          | 0.873   | 0.920   | 0.923   |
> | Llama2 7B ICL Balanced | 0.881   | 0.920   | 0.924   |
> | Llama2 7B ICL Ours     | **0.898**   | **0.922**   | **0.926**   |
>
> These results demonstrate that our method's improvements persist even with balanced demonstrations, suggesting we are addressing a fundamental copying mechanism rather than simply correcting for dataset biases. (We have updated the revised version to include this analysis in Appendix J).
>
> >Minor Reproducibility Concerns: The paper lacks details on certain experimental settings, such as the dataset specifications (e.g., training/validation sizes, number of shots) for the vowel-counting data, as well as the hyperparameter settings used for pruning. Including these details would enhance reproducibility and provide greater clarity on the experimental setup.
>
> We appreciate the reviewer's request to add more details on the experimental settings, to addres this, we have added:
>
> Vowel-Counting Dataset Specifications:
>
> Detection dataset: total of 400 ICL examples.
>
> Each example uses 2-shot format.
>
> Examples are constructed to ensure test answers never appear in prompt responses.
>
> Split: 300 examples for computing neuron importance scores, 100 for validation to determine the optimal pruning configuration.
>
> Words sampled randomly from an English dictionary, filtered to ensure diverse vowel counts (0-9)
>
> We carefully design prompts so that the test words have different vowel counts than example words. (2)
> Hyperparameter settings used for pruning are updated in the revised version (Table 3, Appendix B), including:
> Pruning rates explored per layer.
> Layer selection criteria.

---

> > ### Author Response · Authors · 2024-11-19
> > **Response 3/3**
> >
> > >Limited Exploration of Trade-offs: The paper does not extensively address the potential trade-offs introduced by pruning neurons. Authors only claim that the pruned model never has a drop in performance but what about other side effects? For instance, pruning could theoretically affect the model's performance on tasks where copying might be beneficial (e.g., fact-based retrieval tasks). A more detailed analysis of cases where pruning could have a negative impact would be beneficial.
> >
> > To assess potential tradeoffs introduced by our pruning method, we evaluate the pruned models on perplexity (WikiText) and MMLU benchmark. As shown in the following two tables, our method has minimal impact on these capabilities, while MMLU performance remains stable or even improves in some cases, indicating that our targeted neuron pruning successfully mitigates copying behavior while preserving general capabilities. We have updated the draft to include these results with additional models in Appendix F.
> >
> > Perplexity on WikiText test (lower is better) :
> > |    Model   | Without pruning | With Pruning |
> > |:----------:|:---------------:|:------------:|
> > |  GPT-Small |      **25.188**     |    25.548    |
> > | GPT-Medium |      **18.473**     |    18.590    |
> > | Bloom 560M |     **21.328**     |    21.397    |
> > | Bloom 1.1B |      **16.836**     |    16.883    |
> > |  OPT 125m  |      **26.119**     |    26.432    |
> > |  OPT 350m  |      **20.888**     |    20.940    |
> > |  OPT 1.3B  |      **13.889**    |    13.904    |
> > |  OPT 2.7B  |      **11.757**     |    11.758    |
> > |  Llama2 7b |      **6.5420**    |    6.5490    |
> > | Llama3 78b |      **5.1840**     |    5.8190    |
> >
> > MMLU (higher is better):
> >
> > |       Model      | humanities | social sciences |  stem  |  other |
> > |:----------------:|:----------:|:---------------:|:------:|:------:|
> > |    Gpt2 medium   |   0.2290   |      **0.2427**     | **0.2350** | 0.2128 |
> > | Gpt2 medium+ours |   **0.2427**   |      0.2379     | 0.2294 | **0.2144** |
> > |    Bloom 1.1b    |   0.2482   |      **0.2626**     | 0.2252 | 0.2347 |
> > |  Bloom 1.1b+ours |   **0.2502**   |      0.2620     | **0.2268** | **0.2372** |
> > |     Opt 2.7B     |   **0.2670**   |      0.2398     | **0.2470** | **0.2688** |
> > |  Opt 2.7B + ours |   0.2648   |      **0.2427**     | 0.2457 | 0.2677 |
> > |    Llama 2 7b    |   **0.4329**   |      0.5484     | 0.5297 | **0.3606** |
> > |  Llama 2 7b+ours |   0.4327   |      **0.5491**     | 0.5297 | 0.3603 |
> > |    Llama 3 8b    |   **0.5501**   |      **0.7078**     | 0.7322 | 0.5373 |
> > |  Llama 3 8b+ours |   0.5486   |      0.7048     | **0.7329** | **0.5375** |

---

> > > ### Author Response · Authors · 2024-11-23
> > >
> > > Dear Reviewer ypEw
> > >
> > > As we approach the end of this discussion period, we want to thank you for your thoughtful feedback, which has significantly improved our paper. We believe we’ve addressed most of the concerns raised by you and the other reviewers. In particular, we have incorporated:
> > >
> > > - Adding a balanced prompting baseline.
> > > - Additional implementation details.
> > > - Evaluations across a broader range of tasks (e.g., MMLU, AlpacaEval2) and retrieval challenges like Needle In A Haystack
> > > - Ablation studies exploring the effectiveness of our method beyond the few shot settings.
> > >
> > >
> > >
> > > These experiments demonstrate that our approach consistently mitigates copy bias without compromising performance on tasks where copying is beneficial (e.g., certain retrieval tasks).
> > >
> > > We would appreciate it if you could let us know whether our responses properly address your concerns.

---

> > > > ### Comment · Reviewer_ypEw · 2024-11-24
> > > > **Thank you for details comments!**
> > > >
> > > > I want to thank authors for detailed comments! Authors have adequately addressed all my questions (few-shot setting, and effect of ICL). Authors also provided sufficient justification for the weakness points I raised. I noticed that my review score was among the highest. I have read through other reviewers comments and concerns, but I still think that this paper is interesting and the experiments and analysis are comprehensive. I would like to keep my score.

---

### Official Review · Reviewer_ZJtA · 2024-11-03

**Soundness:** 2
**Presentation:** 2
**Contribution:** 2
**Rating:** 3
**Confidence:** 4

**Summary:**

This paper introduces "copying bias" in large language models (LLMs) in-context learning and presents a method for detecting copying neurons and improves the performance on ICL tasks. The author conducts experiments on ICL tasks and task vectors to verify the claim.

**Strengths:**

1. The author uses illustration to help the reader better understand how the proposed method works.
2. The author considers different ICL tasks for validation.

**Weaknesses:**

1. Lacks some implementation detail: in Section 3.3, what is the size of n? The author assumes that $\hat{y}\notin S_p$ and therefore if n is large, there will be many repetitions in $S_p$ since labels in $S_p$ are the count of vowels, which are integers. Also, does the author explicitly held out an integer so that when constructing $S_p$, the words cannot have the number of vowels equal to this held-out integer?
2. Universality of copying neurons detection: This paper only studies how to detect copy neurons on the vowel counting task. What if this proxy ICL task is switched to other tasks? Will the copying neuron detected still work, and if not (e.g., if all elements in $S_p$ are unique), what makes vowel counting special that it works as a proxy ICL task?
3. Continuation of the previous point: method in 3.3 assumes that the output has single token. This is probably fine for the vowel counting task as single-digits are single-tokens. What if we have other ICL tasks as proxy ICL task and the outputs are multi-tokens?
4. How are errors being calculated? There are two ways: 1) probability-based: all probability summed for labels not equal to the true label and averaged across all inputs; 2) accuracy-based: 1 - (accuracy for the correct output) with greedy decoding.
5. In Figure 2 and 3, it seems that only models up to 1.3B are evaluated. In Figure 2 it seems that a large proportion of error comes from the copying error. However, for larger models (e.g., 8B, 14B) and for the singular to plural task, I suspect that larger model will not have this high copying error if each in-context example is unique. For 1-shot and 2-shot setting, the performance might simply be improved by adding a task description at the very front to eliminate task ambiguity.
6. Pruning the neuron is essentially "training" the model on the ICL task. It would be convincing if the author presents other training-based method as baselines (with a sweep of hyperparameter on the validation set) in Figure 2 and 3.
7. For Figure 5, while there is improvement, the author does not show how much improvement comes from less copying bias. The author has larger models like Llama-3-8B in Figure 4 experiment but there lacks evidence that the pruning decreases the copying error.

Minor issues:
1. Example in the introduction can be ambiguous. it could also be the case to count the number of words and this case the correct answer for "Florida" is 1.
2. Line 063-064: the first quotation mark of "Copying Neurons." is not correct.

**Questions:**

Please see the weakness section above where I asked some questions.

---

> ### Author Response · Authors · 2024-11-19
> **Response 1/4**
>
> We are grateful for your detailed review, which has guided us in improving our paper.
>
> >Lacks some implementation detail: in Section 3.3, what is the size of n? The author assumes that y^∉Sp and therefore if n is large, there will be many repetitions in Sp since labels in Sp are the count of vowels, which are integers. Also, does the author explicitly held out an integer so that when constructing Sp, the words cannot have the number of vowels equal to this held-out integer?
>
> In our implementation, we used n=2 examples per prompt, as this setting more clearly exposes the copying tendency in models, making it ideal for detecting the relevant neurons. When constructing prompts, we ensure each prompt's examples (Sp) contain distinct vowel counts, and the target word's vowel count (ŷ) differs from those in Sp, sampling from words with 1-8 vowels.
>
> >Universality of copying neurons detection: This paper only studies how to detect copy neurons on the vowel counting task. What if this proxy ICL task is switched to other tasks? Will the copying neuron detected still work, and if not (e.g., if all elements in are unique), what makes vowel counting special that it works as a proxy ICL task?
>
> Our results show that neurons identified using any of these tasks led to similar improvements in performance across the broader set of evaluation tasks. The key requirement for an effective proxy task appears to be that it: 1) has clear, deterministic rules that require pattern recognition rather than memorization, 2) can elicit copying behavior from the model, and 3) has a sufficiently constrained output space to detect copying errors reliably. While vowel counting meets these criteria particularly well and is easy to generate synthetic data for, it is not uniquely special - other structured tasks with similar properties could serve as effective proxies for copying neuron detection. To verify this, we conducted additional experiments using different proxy ICL tasks beyond vowel counting, mainly the two linguistics tasks: linguistic antonyms (happy → sad) and knowledge person language (Muhammad → Islam).
>
> The results below are averaged across the following tasks : (linguistic present simple to gerund, linguistic present simple to past simple, linguistic present simple to past perfect, linguistic singular to plural) over 5 different seeds. In each cell of the table, on the left side, we report the result of the unpruned version of the model, and on the right side, we show the result of the pruned version using our method. We have updated the draft to include this analysis in Appendix I.
>
> For the vowel counting task:
>
> | Model       | 1 Shot    | 2 Shot    | 3 Shot    | 4 Shot    |
> |-------------|-----------|-----------|-----------|-----------|
> |  GPT2 Small | 0.16,**0.25** | 0.31,**0.44** | 0.32,**0.44** | 0.37,**0.47** |
> | GPT2 Medium | 0.24,**0.30** | 0.38,**0.45** | 0.44,**0.52** | 0.49,**0.58** |
> |  Bloom 560M | 0.35,**0.42** | 0.42,**0.53** | 0.48,**0.59** | 0.50,**0.59** |
> |   OPT 1.3B  | 0.21,**0.24** | 0.30,**0.42** | 0.42,**0.52** | 0.50,**0.59** |
>
> For the linguistic antonyms task:
>
> |    Model    |   1 Shot  |   2 Shot  |   3 Shot  |   4 Shot  |
> |:-----------:|:---------:|:---------:|:---------:|:---------:|
> |  GPT2 Small | 0.16,**0.21** | 0.31,**0.35** | 0.32,**0.37** | 0.37,**0.42** |
> | GPT2 Medium | 0.24,**0.30** | 0.38,**0.47** | 0.44,**0.52** | 0.50,**0.58** |
> |  Bloom 560M | 0.35,**0.37** | 0.42,**0.48** | 0.48,**0.53** | 0.50,**0.58** |
> |   OPT 1.3B  | 0.21,**0.23** | 0.30,**0.32** | 0.42,**0.44** | 0.50,**0.52** |
>
> For the knowledge person language task:
>
> |    Model    |   1 Shot  |   2 Shot  |   3 Shot  |   4 Shot  |
> |:-----------:|:---------:|:---------:|:---------:|:---------:|
> |  GPT2 Small | 0.16,**0.21** | 0.31,**0.35** | 0.31,**0.37** | 0.36,**0.42** |
> | GPT2 Medium | 0.24,**0.30** | 0.38,**0.47** | 0.44,**0.52** | 0.50,**0.58** |
> |  Bloom 560M | 0.35,**0.37** | 0.42,**0.48** | 0.48,**0.53** | 0.50,**0.58** |
> |   OPT 1.3B  | 0.21,**0.23** | 0.30,**0.32** | 0.42,**0.44** | 0.50,**0.52** |
>
>
> Our experiments with different proxy tasks (vowel counting, linguistic antonyms, and knowledge person language) demonstrate comparable improvements across models and shot settings, with vowel counting showing slightly stronger gains, suggesting that while multiple structured tasks can effectively identify copying neurons, tasks with clear deterministic rules and constrained output spaces like vowel counting may be particularly well-suited for this purpose.

---

> ### Author Response · Authors · 2024-11-19
> **Response 2/4**
>
> > Continuation of the previous point: method in 3.3 assumes that the output has single token. This is probably fine for the vowel counting task as single-digits are single-tokens. What if we have other ICL tasks as proxy ICL task and the outputs are multi-tokens?
>
> We demonstrate that our method, despite using single-token vowel counting as a proxy task, effectively generalizes to multi-token outputs. As empirical evidence, we show consistent improvements on the country-currency task where outputs are typically multi-token (e.g., "Benin" → "CFA Franc (XOF)") and national parks ("Badlands National Park": "South Dakota")., indicating that neurons identified through single-token analysis still effectively mitigate copying bias in multi-token scenarios. In each cell of the table, on the left side, we report the result of the unpruned version of the model, and on the right side, we show the result of the pruned version using our method.
>
>  national parks:
>
> |    Model   |   1 Shot  |   2 Shot  |   3 Shot   |   4 Shot  |
> |:----------:|:---------:|:---------:|:----------:|:---------:|
> | Bloom 560M | 0.03,**0.12** | 0.06,**0.15** | 0.07,**0.19** | 0.07,**0.20** |
> | GPT2-Small | 0.04,**0.09** | 0.09,**0.14** |  0.10,**0.12** | 0.10,**0.14** |
>
>  Country-currency:
>
> |    Model   |   1 Shot  |   2 Shot  |   3 Shot  |   4 Shot  |
> |:----------:|:---------:|:---------:|:---------:|:---------:|
> | Bloom 560M | 0.06,**0.10** | 0.12,**0.15** | 0.12,**0.17** | 0.11,**0.15** |
> | GPT2-Small | 0.06,**0.10** | 0.08,**0.12** | 0.09,**0.12** | 0.09,**0.13** |
>
> Our detailed analysis of multi-token tasks (national parks and country-currency pairs) shows that our method improves performance specifically on multi-token predictions. For example, on the country-currency task, improvements are observed on multi-token currencies like 'New Zealand Dollar'  and 'Saudi Riyal'. This demonstrates that while our proxy task uses single-token outputs for neuron identification, the benefits effectively generalize to multi-token prediction scenarios, suggesting the identified copying neurons play a role in sequence-level prediction behaviors.
>
> While our current implementation uses single-token proxy tasks for simplicity, the method could be naturally extended to explicitly handle multi-token outputs by aggregating prediction shifts across the sequence: ∆L = Σt(l_v^t - l_u^t), where t indexes token positions. This approach aligns with standard practices in applying integrated gradients to generation tasks. However, our results suggest that such an extension may not be necessary in practice, as the current approach already generalizes well to multi-token settings. We have updated the revised version to include these results in Appendix E.
>
> > How are errors being calculated? There are two ways: 1) probability-based: all probability summed for labels not equal to the true label and averaged across all inputs; 2) accuracy-based: 1 - (accuracy for the correct output) with greedy decoding.
>
> In our evaluation, we use accuracy-based error calculation with greedy decoding (method 2). A prediction is counted as an error if the model's output with the highest probability doesn't match the ground truth. For multi-token outputs, we require an exact match of the complete sequence. While this approach provides a straightforward way to measure performance, we note that it may be conservative in cases where multiple valid answers exist.

---

> ### Author Response · Authors · 2024-11-19
> **Response 3/4**
>
> > In Figure 2 and 3, it seems that only models up to 1.3B are evaluated. In Figure 2 it seems that a large proportion of error comes from the copying error. However, for larger models (e.g., 8B, 14B) and for the singular to plural task, I suspect that larger model will not have this high copying error if each in-context example is unique. For 1-shot and 2-shot setting, the performance might simply be improved by adding a task description at the very front to eliminate task ambiguity.
>
> Following the review, we have experimented with using task descriptions. In our experiments, we compare three settings: (1) standard ICL without task descriptions (baseline), (2) ICL with added task descriptions explaining the required transformation (e.g., "Convert verbs from present to gerund form"), and (3) our pruning method. While adding task descriptions improves over the baseline, our pruning method achieves substantially better results across all settings , suggesting that copying bias persists even with explicit task specifications and requires targeted intervention through neuron pruning. Interestingly, combining task descriptions with our pruning method (TD+Pruning) yields the best performance across all models and shot settings, and maintains consistent gains even for larger models like Llama3 8B, suggesting these approaches are complementary in addressing different aspects of the copying bias problem.
> The results below are averaged across the following tasks : (linguistic present simple to gerund, linguistic present simple to past simple, linguistic present simple to past perfect, linguistic singular to plural) over 5 different seeds. (TD refers to adding Task Description).  We have updated the revised version to include this analysis, at Appendix J.
>
> | Model                      | 1 Shot | 2 Shot | 3 Shot | 4 Shot |
> |----------------------------|--------|--------|--------|--------|
> | GPT2-Small+ICL             | 0.16   | 0.31   | 0.32   | 0.37   |
> | GPT2-Small+ICL+TD          | 0.19   | 0.35   | 0.37   | 0.41   |
> | GPT2-Small+ICL+Pruning     | 0.25   | 0.44   | 0.44   | 0.47   |
> | GPT2-Small+ICL+TD+Pruning  | **0.27**   |  **0.45**  | **0.46**   | **0.50**   |
> | Bloom 560M+ICL             | 0.35   | 0.42   | 0.48   | 0.50   |
> | Bloom 560M+ICL+TD          | 0.37   | 0.46   | 0.43   | 0.55   |
> | Bloom 560M+ICL+Pruning     | 0.42   | **0.53**   | 0.59   | 0.59   |
> | Bloom 560M+ICL+TD+Pruning  | **0.44**   | **0.53**   | **0.61**   | **0.60**  |
> | Llama3 8B+ICL              | 0.68   | 0.85   | 0.88   | 0.89   |
> | Llama3 8B+ICL+TD           | 0.70   | 0.88   | 0.89   | **0.90**   |
> | Llama3 8B+ICL+Pruning      | **0.73**   | 0.89   | **0.90**   | **0.90**   |
> | Llama3 8B+ICL+TD+Pruning   | **0.73**   | **0.90**   | **0.90**   | **0.90**   |
>
> > Pruning the neuron is essentially "training" the model on the ICL task. It would be convincing if the author presents other training-based method as baselines (with a sweep of hyperparameter on the validation set) in Figure 2 and 3.
>
> We appreciate the reviewer's suggestion to compare our pruning approach with training-based methods. To address this, we conducted experiments comparing our pruning method with fine-tuning on the vowel mapping task. Specifically, we fine-tuned models using the same synthetic vowel mapping dataset used for neuron identification, performing a thorough hyperparameter sweep (learning rates: [1e-5, 1e-4, 1e-3], batch sizes: [4, 8, 16], epochs: [1, 3, 5]) and selecting the best configuration using the validation set.
> Results are averaged across all of the linguistic tasks for 5 different seeds, the left number in each cell presents the baseline model without any modification, and the right number in each cell presents the result of the finetuned/pruned model. We have updated the revised version to include this analysis in Appendix J.
>
> |        Model       |   1 Shot  |   2 Shot  |   3 Shot  |   4 Shot  |
> |:------------------:|:---------:|:---------:|:---------:|:---------:|
> | Bloom 560M Fintued | 0.35,0.34 | 0.42,0.41 | 0.48,0.45 | 0.50,0.50 |
> |   Bloom 560M Ours  | 0.35,**0.42** | 0.42,**0.53** | 0.48,**0.59** | 0.50,**0.59** |
> | GPT2-Small Fintued | 0.16,0.14 | 0.31,0.31 | 0.32,0.32 | 0.37,0.35 |
> |   GPT2-Small Ours  | 0.16,**0.25** | 0.31,**0.44** | 0.32,**0.44** | 0.37,**0.47** |
> | OPT-1.3B Finetuned | 0.21,0.20 | 0.30,0.30 | 0.42,0.42 | 0.50,0.50 |
> |    OPT-1.3B Ours   | 0.21,**0.24** | 0.30,**0.42** | 0.42,**0.52** | 0.50,**0.59** |
>
> These results demonstrate that finetuning on the vowel mapping task often degrades or maintains baseline performance. This may arise from an overfitting to the synthetic task caused by the finetuning process. In contrast, as shown in our paper our pruning method consistently yields substantial improvements across all models and shot settings.

---

> ### Author Response · Authors · 2024-11-19
> **Response 4/4**
>
> > For Figure 5, while there is improvement, the author does not show how much improvement comes from less copying bias. The author has larger models like Llama-3-8B in Figure 4 experiment but there lacks evidence that the pruning decreases the copying error.
>
> Task vectors, as introduced by Hendel et al., represent the model's learned understanding of a task from in-context examples - essentially capturing the transformation needed to solve the task. In Figure 5, we compare two types of task vectors: "Task-Vectors" refers to task vectors obtained from the original unpruned model which exhibits more copying errors, while "Task-Vectors-Pruned" refers to task vectors obtained from our pruned model which reduces these copying errors. The results demonstrate that Task-Vectors-Pruned significantly outperforms Task-Vectors across the different shots and tasks. providing strong evidence that our pruning strategy enhances the model's ability to distill task-relevant information into high-quality task vectors.
>
> In the following, we show the percentage of the copying errors among the total errors for the standard ICL method compared to the case with pruning the copying neurons detected by our method over the BBH ICL task. Evidently, our method indeed decreases the copying error also on larger models like Llama3 8B (We have updated the revised version to include this analysis in Appendix C):
>
> | Model              | 1 Shot | 2 Shot | 3 Shot | 4 Shot |
> |--------------------|--------|--------|--------|--------|
> | Llama 3 8B ICL        | 10%    | 16%    | 19%    | 24% |
> | Llama 3 8B ICL + Ours | 4%     | 10%    | 12%    | 15% |
>
> > Example in the introduction can be ambiguous. it could also be the case to count the number of words and this case the correct answer for "Florida" is 1.
>
> Thank you for this comment. Following this concern, we have replaced this example in the revised version.
>
> > Line 063-064: the first quotation mark of "Copying Neurons." is not correct.
> Thank you for pointing this. Following this concern, we have fixed it along with many other typos.

---

> ### Author Response · Authors · 2024-11-23
>
> Dear Reviewer ZJtA
>
> As we approach the end of this discussion period, we want to thank you for your thoughtful feedback, which has significantly improved our paper. We believe we’ve addressed most of the concerns raised by you and the other reviewers. In particular, we have incorporated:
>
> - Additional implementation details.
> - A fine-tuning baseline.
> - A Task Description baseline.
> - Evaluations across a broader range of tasks (e.g., MMLU, AlpacaEval2) and retrieval challenges like Needle IN A Haystack
> - Experiments demonstrating stability to the choice of the proxy task
> - Experiments demonstrating the effectiveness of our method over multi-token output tasks.
>
>
> These experiments demonstrate that our approach consistently mitigates copy bias without compromising performance on tasks where copying is beneficial (e.g., certain retrieval tasks).
>
> We would appreciate it if you could let us know whether our responses properly address your concerns.

---

> > ### Comment · Area_Chair_PVFS · 2024-11-24
> > **discussion**
> >
> > Dear Reviewer ZJtA -- could you please look at the authors' rebuttal and acknowledge that you've read it?  Also, if you have any further questions for the authors, please let them know ASAP.

---

> ### Comment · Reviewer_ZJtA · 2024-12-02
>
> Sorry for the late response. I thank the author for their reply. I think the paper has improved, and the new experiments added show the effectiveness on simple tasks like vowel counting and on smaller models. However, I still have some concerns about the paper.
> * This paper studies copying bias of ICL. The paper can fall into one of the two categories, each with different criterion to evaluate the contribution of the paper:
>   * (1) "Methodology" paper, where the contribution is evaluated based on how applicable / useful the new methodology can be, or
>   * (2) "Finding" paper, where a detailed analysis need to be conducted to study copying bias (e.g., mechanistically; Reviewer 4ESn also mentions this).
> * It seems that the paper falls more into a "methodology" paper as the author claims that "consistently improves performance across a diverse set of ICL tasks". However, I am not sure this claim is fully supported with current evidence in the experiments and the applicability is limited.
>   * As I mentioned in the review, "I suspect that larger model will not have this high copying error if each in-context example is unique". While the author provides table for experiments in the rebuttal, looking at the results of Llama-3 8B, the improvement is quite marginal (also I'm not sure how much percentage is from mitigating copying bias and whether there are cases where, after pruning, get worse).
>   * While one can argue that there is 3 percent improvement in the 1-shot setting, the performance in 2-/3-/4- shot setting is very close to (or the same as) the TD case. The additional cost from adding additional ICL samples is small (so one can choose to switch to 4-shot ICL). Also, there is an extra cost of applying the pruning on the model.
>   * The tasks are relatively simple.
>
> For these reasons, I will keep the current score. I would encourage the author to do a more detailed analysis on copying bias.

---

> ### Author Response · Authors · 2024-12-02
>
> Dear Reviewer ZJtA,
>
> Thank you for your detailed feedback and for engaging with our rebuttal and additional experiments.
>
> We would like to respond to the concerns you raised. We believe our contribution bridges both categories you outlined – Findings and Methodology. On the methodological side, we propose an approach to mitigate copying bias effectively. Concurrently, we aim to provide insights into the behavior of ICL approaches regarding copying bias. While "mechanistic" might not correctly capture the nature of our findings, we do establish a connection between copying bias and inaccuracies in task vector predictions, demonstrating that pruning improves inference by addressing these inaccuracies. Furthermore, we find it compelling that this shortcut is largely task-agnostic; that is, detecting neurons on a single task allows us to influence other tasks in a functionally isolated and effective manner.
>
> On the methodological front, we highlight the consistent reduction in copying error.  While larger models and multi-shot ICL settings are inherently more accurate, leading to smaller absolute reductions, the relative impact in reduction of copying of our approach remains substantial, and the proportion of copy errors remains high or even increases. For instance, as highlighted in Table 5 (Appendix C),: 15% of errors in LLaMA3 (8B) with 4-Shot ICL on BBH are copy errors, compared to only 4% with 1-Shot ICL. This highlights the continued relevance of our approach, even in advanced settings. Furthermore, the pruning process is computationally inexpensive and only needs to be performed once, making the method practical for real-world scenarios.
> We also wish to point out that these new concerns regarding the magnitude of improvement and the positioning of the paper were not part of your initial review. Nevertheless, we have made extensive revisions to address all 7 weaknesses you originally highlighted. The paper has substantially improved in clarity and experimental support as a result of your valuable input.
>
> 1. Lack of implementation details: We’ve added detailed explanations (see lines 185 - 193).
> 2. Universality of copy neuron detection: We’ve demonstrated stability with respect to the choice of the proxy task (see Appendix I).
> 3. Only single-token tasks: The model was already evaluated on multi-token tasks. We clarified this and emphasized the improvements in multi-token setups (see Appendix E).
> 4. Error calculation unclear: We clarified that the standard metric, exact match is used
> 5. Results in Fig 2 and 3 limited to 1.3B model & Task Description: We incorporated results with LLaMA 3 (8B model) (see Appendix J).
> 6. Comparison with standard training-based baselines missing: We added these comparisons and showed that the baseline performs poorly (see Appendix J).
> 7. Source of improvement (reduced copying bias): We added a new table confirming that the improvement is indeed due to reduced copying bias (Appendix C).
>
> We greatly appreciate your careful evaluation and hope this response helps clarify the significance of our work and its positioning. We respectfully ask you to reconsider the score in light of the substantial improvements and alignment with your original feedback.
>
> Thank you once again for your time and effort.
>
> Sincerely,
>
> The Authors

---

### Official Review · Reviewer_9Q7b · 2024-11-04

**Soundness:** 3
**Presentation:** 3
**Contribution:** 2
**Rating:** 6
**Confidence:** 4

**Summary:**

The paper presents a novel approach to addressing a key issue in large language models (LLMs) during In-Context Learning (ICL)—the tendency to copy answers from provided examples, known as “copying bias.” Instead of learning and generalizing from the patterns in the context, models often replicate the examples, especially in few-shot learning settings. To tackle this, the authors propose using Integrated Gradients (IG) to identify neurons responsible for this copying behavior. By pruning these “copying neurons,” the model’s ability to reason and generalize is improved.

The method is task-agnostic and applicable across various LLM architectures, including Transformer-based models like GPT, Bloom, OPT, and LLaMA, as well as State-Space Models such as Mamba. Extensive experiments across different tasks show that neuron pruning significantly reduces copying errors, enhancing overall task performance. The paper also demonstrates that this pruning improves the quality of task vectors, leading to better task recognition and generalization.

**Strengths:**

This paper successfully implemented neural pruning into tackling the copy bias in ICL. The proposed IG method with pruning strategy are relatively reasonable to solve the copy bias problem. This is a significant contribution as previous work primarily focused on prompt engineering and calibration methods without directly addressing the internal model dynamics causing copying errors.

Also, the proposed method is versatile, as it can be applied across LLM architectures, including both Transformersand SSM, without needing task-specific modifications. This general applicability makes the method highly practical for a wide range of ICL tasks and model types. It provides “out-of-the-box” improvements without requiring model retraining or access to large amounts of task-specific data, making it an efficient post-processing step in LLM deployment pipelines.

The authors use a vowel-counting task as a proxy for detecting copying errors, where language models are prone to making copying mistakes. The proxy ICL dataset generation method is clear are easy to follow. The diverse range of tasks helps to demonstrate the robustness and general applicability

**Weaknesses:**

The primary weakness of this paper is the lack of a comprehensive evaluation of the model’s capabilities after pruning. While the method effectively mitigates copy bias, this paper presents more of a case study rather than demonstrating its real-world applicability. Pruning techniques generally have an impact on the overall performance of the model, and as such, it is critical for the authors to provide results showing how the pruned model performs on downstream tasks. Unfortunately, this paper does not offer that kind of evaluation, which is a necessary component to fully understand the implications of the pruning approach on the model’s general capabilities.

Another issue with the paper is the limited comparison with baselines. The authors provide several ICL-related baselines, but they seem to operate under the assumption that copy bias can only be addressed with ICL-focused solutions. The paper would benefit from a broader exploration of the problem, including a more detailed definition of copy bias and an examination of its resolution under different types of models, such as base and instruct models. Notably, the authors did not experiment with instruct models, which may behave very differently, especially with methods like APE that are known to improve prompt-following capabilities. Additionally, it would be valuable to explore whether fine-tuning or data engineering, such as modifying the SFT dataset or providing higher-quality alignment data, could address the issue. If data engineering alone could solve the problem, one must question the necessity of introducing an additional, complex pruning process, particularly for large models where pruning is a non-trivial engineering task.

**Questions:**

- How does the pruned model perform on downstream tasks, such as reasoning, math, long-context understanding, instruction following, and safety? While it may not be necessary to conduct an exhaustive evaluation due to computational resource limitations, at the very least, relevant experiments should be included.

- Does it matter if the model is a base model or an instruct model? For example, if Bloom-560M is turned into an instruct model or fine-tuned with DPO, will the experimental results change or remain the same?

- Can data engineering help solve this problem? If it can, why do we still need an additional step to prune the model?

---

> ### Author Response · Authors · 2024-11-19
>
> Thank you for taking the time to provide such insightful feedback.
>
> >How does the pruned model perform on downstream tasks, such as reasoning, math, long-context understanding, instruction following, and safety? While it may not be necessary to conduct an exhaustive evaluation due to computational resource limitations, at the very least, relevant experiments should be included.
>
> To assess potential tradeoffs introduced by our pruning method, we evaluate the pruned models on perplexity (WikiText) and MMLU benchmark. As shown in the following two tables, our method has minimal impact on these capabilities, while MMLU performance remains stable or even improves in some cases, indicating that our targeted neuron pruning successfully mitigates copying behavior while preserving general capabilities. We have updated the draft to include these results with additional models in Appendix F.
>
> Perplexity on WikiText test (lower is better) :
> |    Model   | Without pruning | With Pruning |
> |:----------:|:---------------:|:------------:|
> |  GPT-Small |      **25.188**     |    25.548    |
> | GPT-Medium |      **18.473**     |    18.590    |
> | Bloom 560M |     **21.328**     |    21.397    |
> | Bloom 1.1B |      **16.836**     |    16.883    |
> |  OPT 125m  |      **26.119**     |    26.432    |
> |  OPT 350m  |      **20.888**     |    20.940    |
> |  OPT 1.3B  |      **13.889**    |    13.904    |
> |  OPT 2.7B  |      **11.757**     |    11.758    |
> |  Llama2 7b |      **6.5420**    |    6.5490    |
> | Llama3 78b |      **5.1840**     |    5.8190    |
>
> MMLU (higher is better):
>
> |       Model      | humanities | social sciences |  stem  |  other |
> |:----------------:|:----------:|:---------------:|:------:|:------:|
> |    Gpt2 medium   |   0.2290   |      **0.2427**     | **0.2350** | 0.2128 |
> | Gpt2 medium+ours |   **0.2427**   |      0.2379     | 0.2294 | **0.2144** |
> |    Bloom 1.1b    |   0.2482   |      **0.2626**     | 0.2252 | 0.2347 |
> |  Bloom 1.1b+ours |   **0.2502**   |      0.2620     | **0.2268** | **0.2372** |
> |     Opt 2.7B     |   **0.2670**   |      0.2398     | **0.2470** | **0.2688** |
> |  Opt 2.7B + ours |   0.2648   |      **0.2427**     | 0.2457 | 0.2677 |
> |    Llama 2 7b    |   **0.4329**   |      0.5484     | 0.5297 | **0.3606** |
> |  Llama 2 7b+ours |   0.4327   |      **0.5491**     | 0.5297 | 0.3603 |
> |    Llama 3 8b    |   **0.5501**   |      **0.7078**     | 0.7322 | 0.5373 |
> |  Llama 3 8b+ours |   0.5486   |      0.7048     | **0.7329** | **0.5375** |
>
> > Does it matter if the model is a base model or an instruct model? For example, if Bloom-560M is turned into an instruct model or fine-tuned with DPO, will the experimental results change or remain the same?
>
> Following the reviewer request, we report below the results of OPT1.3B base and OPT1.3B Instruct [1]:
> These results are averaged across the following tasks : (linguistic present simple to gerund, linguistic present simple to past simple, linguistic present simple to past perfect, linguistic_singular_plural, linguistic_antonyms), and across 5 different seeds. (we have updated the draft in Appendix J to include this analysis).
>
> |       Model       |   1 Shot  |   2 Shot  |   3 Shot  |   4 Shot  |
> |:-----------------:|:---------:|:---------:|:---------:|:---------:|
> |      OPT 1.3B     | 0.28,**0.32** | 0.38,**0.48** | 0.49,**0.57** | 0.55,**0.62** |
> | OPT 1.3B instruct | 0.27,**0.32** | 0.39,**0.46** | 0.51,**0.57** | 0.55,**0.64** |
>
> In each cell of the table, on the left side, we report the result of the unpruned version of the model, and on the right side, we show the result of the pruned version using our method.
> The results demonstrate that our pruning method effectively improves performance for both base and instruction-tuned models, with similar relative improvements observed across different shot settings, suggesting that copying bias and the effectiveness of our mitigation strategy persist regardless of the model's fine-tuning regime.
>
> [1] : https://huggingface.co/facebook/opt-iml-1.3b
>
> >Can data engineering help solve this problem? If it can, why do we still need an additional step to prune the model?
>
> We agree that an alternative approach could involve incorporating data or a learning objective designed to prevent this reasoning shortcut from emerging, although this may prove to be nontrivial. One of the key contributions of our submission is precisely the identification and demonstration of the significance of this shortcut, which may be then tackled from various angles. In any case, we believe, our approach remains valuable because many existing models exhibit this behavior. Furthermore, we believe our method serves as a compelling example of how neuron-level editing can effectively discourage shortcuts, with potential applications in other settings — for instance, addressing the ‘lost in the middle’ issue in long-context generation.

---

> ### Comment · Reviewer_9Q7b · 2024-11-19
>
> Thank you very much for the author’s response. I believe that the experiments you provided are quite comprehensive. However, I still have a few questions that I would appreciate your response to:
> - In the reply addressing Q1, the author presented results related to perplexity (PPL) and downstream tasks (MMLU). On WikiText, it appears that the PPL of the model has shown a certain degree of degradation, yet the MMLU score of the model has maintained its original level to some extent. I am somewhat skeptical that the MMLU dataset is “favored” by pruning. For example, in earlier pruning works like ShortGPT or The Unreasonable Ineffectiveness of the Deeper Layers, MMLU has often been a dataset that preserves relatively good performance. My hypothesis is that MMLU-related tasks are among the least sensitive to pruning, whereas other tasks, such as associative recall or safety issues, seem to be less effectively addressed by pruning. Therefore, I suggest the author also provide experiments related to these types of tasks (considering that recall is a commonly used LLM application scenario, such as locating and copying large sections of text from PDFs).
> - Thank you for your response to Q2, which has partially resolved my doubts. I recommend testing with datasets such as AlpacaEval2. Such datasets seem to be closer to typical usage scenarios.
> - Thank you for addressing Q3. However, I am still very interested in whether data engineering can address this issue. Data engineering appears to be a simpler approach. For example, most commercially available LLMs have not undergone pruning.

---

> ### Author Response · Authors · 2024-11-21
>
> Thank you very much for the prompt response.
>
> >  I recommend testing with datasets such as AlpacaEval2.
>
> We tested on AlpacaEval2 (https://github.com/tatsu-lab/alpaca_eval). Due to computational constraints, we evaluated on half of the test set. For reference, the original Llama2's performance on the full test set achieves 5.4% LC win rate and 5.0% win rate. On our half-set evaluation, the unpruned Llama2 achieves 4.84% LC win rate and 2.69% win rate, while our pruned version improves to 5.26% LC win rate and 3.21% win rate.  This (along with results on MMLU) suggests that our pruning method preserves the model’s ability to handle both general language tasks. (The results of this experiment are updated in the revised version).
>
>
> | Model                            | LC Win Rate | Win Rate |
> |----------------------------------|-------------|----------|
> | Unpruned Llama2 (half test set)  | 4.84%       | 2.69%    |
> | Pruned Llama2 (half test set)    | 5.26%       | 3.21%    |
>
> > Therefore, I suggest the author also provide experiments related to these types of tasks (considering that recall is a commonly used LLM application scenario, such as locating and copying large sections of text from PDFs).
>
> We considered the Needle-In-A-Haystack benchmark (https://github.com/gkamradt/LLMTest_NeedleInAHaystack), which evaluates the retrieval capabilities of LLMs. This benchmark asks the LLM to locate a specified key within a long context and accurately reproduce the corresponding value from the identified location. The results (reported in the revised version at Appendix F.2 and below) show only a slight difference in performance between pruned and unpruned versions of the tested model across varying context lengths. The high consistency across context and depth length demonstrates that while our pruning effectively mitigates copying bias in ICL settings, it does not compromise any model’s fundamental ability to process and copy information from long contexts.
>
> In the following table we show the averaged fact retrieval scores (in percentages) for pruned and unpruned Llama2 model for the needle-in-a-haystack benchmark using PaulGrahamEssays as a context ‘high-stack’, and the following sentence as the ‘needle’ : “The best thing to do in San Francisco is eat a sandwich and sit in Dolores Park on a sunny day.” tested at different context window sizes ranging from 1000 to 6000 tokens (shown at even intervals), where higher scores indicate better performance at finding the relevant information in the given context length.
>
> Llama-2 7B:
>
> | Context Len | 1231 | 1692 | 2154 | 2615 | 3077  | 3538  | 4000  | 4462  | 4923 | 5385 |
> |-------------|------|------|------|------|-------|-------|-------|-------|------|------|
> | unpruned    | 100% | 100% | 100% | 100% | 92.5% | 85.0% | 94.2% | 13.3% | 0.0% | 0.8% |
> | pruned      | 100% | 100% | 100% | 100% | 85.8% | 90.0% | 91.7% | 13.3% | 0.0% | 0.0% |
>
> >  I am still very interested in whether data engineering can address this issue.
>
> While data engineering during pretraining or alignment could in principle help address copying bias, reliably testing this hypothesis would require extensive modifications to training data and procedures - a significant undertaking that warrants its own dedicated investigation. However, we have concretely tested the post-hoc data engineering approach through fine-tuning experiments on our proxy task data (see Table 18, Appendix J). Our results show that fine-tuning fails to mitigate copying bias effectively, with performance actually degrading in some cases compared to the base model. This suggests that simply retraining on carefully engineered data may not be sufficient to address the fundamental copying mechanisms in these models. We welcome suggestions for specific data engineering approaches that could be empirically evaluated against our method. If we won’t be able to implement them during the discussion period, we will at least try to incorporate them in the final version of the paper.

---

> > ### Comment · Reviewer_9Q7b · 2024-11-21
> >
> > **On the Needle-In-A-Haystack Experiment**: I believe the authors should adopt more rigorous experimental setups. From the table provided, this experiment appears overly simplistic (with no details on how many trials were conducted or whether results were averaged). Additionally, the reported accuracy seems to differ from that in some related works, such as this paper https://arxiv.org/pdf/2310.03025.
> >
> > **On the Data Engineering Experiment**: I understand that discussing experiments related to data engineering during the rebuttal period, or even in the entire paper, is unrealistic. This issue will not affect my personal scoring in any way. I am raising this purely out of academic interest and hope to engage in a discussion with the authors. I believe addressing this could contribute to the completeness of the paper.

---

> > > ### Author Response · Authors · 2024-11-23
> > >
> > > > Experiment on Alpaca Eval2: This is the experiment that confuses me the most. I am curious about how the authors managed to achieve such a significant improvement in the model’s performance on this dataset through neural pruning. My initial motivation for testing this issue was that if neural pruning could maintain the model’s capabilities without loss (or with minimal loss, which could be mitigated through fine-tuning), it would prove the reasonableness of this method. However, what is most intriguing is that this method not only maintained but also enhanced the model’s capabilities, achieving an improvement close to a percentage point.
> > >
> > >
> > > In AlpacaEval2, the evaluation compares model responses directly against GPT-4’s, which can make it less reliable for smaller models (like the LLaMA 2 7B-based models we reported on). Given this, we believe the observed 0.5-point difference in win rate (4.8% for the original model vs. 5.3% for the pruned version) is small enough to be interpreted as no meaningful difference.
> > >
> > > To make this even more reliable and reconfirm that pruning preserves model capabilities, we conducted pairwise comparisons between the pruned and unpruned models. Here, the unpruned model’s outputs served as a reference, and GPT-4 Turbo was still used as the evaluator. The results showed a win rate of 51.49% for the pruned model (i.e. essentially at parity with the unpruned model).
> > >
> > > These near-parity results -  close to 50% - strongly support the conclusion that our pruning method effectively preserves the model’s capabilities. The key takeaway isn’t the small improvement but the consistent performance between pruned and unpruned models, as demonstrated by both the standard evaluation and the pairwise comparison.
> > >
> > > Thank you again for engaging with this discussion — we truly appreciate your insights!
> > >
> > > > Also, is the "LLaMA-2" model a base one? I think the authors should use LLaMA-2 instruct version.
> > >
> > > > On the Needle-In-A-Haystack Experiment: I believe the authors should adopt more rigorous experimental setups. From the table provided, this experiment appears overly simplistic (with no details on how many trials were conducted or whether results were averaged). Additionally, the reported accuracy seems to differ from that in some related works, such as this paper https://arxiv.org/pdf/2310.03025.
> > >
> > > Thank you for your thoughtful comments.
> > >
> > > Following the reviewer request on adopting more rigorous experimental setups, we have adopted the RULER benchmark (https://arxiv.org/abs/2404.06654). RULER is a comprehensive benchmark that expands beyond the simple needle-in-haystack scenario we reported previously. We have used LLaMA-3.1-Instruct (https://huggingface.co/meta-llama/Llama-3.1-8B-Instruct) as a model to test. In the following table we report the results for both the unpruned version of the model and the pruned version of the model:
> > >
> > > Our goal in this experiments is to demonstrate that mitigating copy-bias using our pruning method does not compromise the model ability to retrieve/copy content from context, thus - and also due to computational and time constraints - we selected tasks which are testing these abilities: “niiah_single_1"  "niah_single_2"  "niah_single_3"  "niah_multikey_1"  "niah_multikey_2" "niah_multikey_3"  "niah_multivalue"  "niah_multiquery"  only:
> > > LLaMA 3.1 Instruct:
> > >
> > > |   Model  | Length |   4K  |   8K  |  16K  |  32K  |  64K  |  128K |  AVG  |
> > > |:--------:|:------:|:-----:|:-----:|:-----:|:-----:|:-----:|:-----:|:-----:|
> > > | Unpruned |  128K  | 99.9% | 99.9% | **99.8%** | **99.6%** | **98.7%** | **92.6%** | **98.4%** |
> > > |  Pruned  |  128K  | 99.9% | 99.9% | 99.7% | 99.4% | 98.4% | 92.1% | 98.2% |
> > >
> > > The results are very similar for both models, reassuring us that pruning does not compromise retrieval ability.
> > >
> > > > On the Data Engineering Experiment: I understand that discussing experiments related to data engineering during the rebuttal period, or even in the entire paper, is unrealistic. This issue will not affect my personal scoring in any way. I am raising this purely out of academic interest and hope to engage in a discussion with the authors. I believe addressing this could contribute to the completeness of the paper.
> > >
> > > We plan to expand our discussion in the next revision to address two key aspects. First, we will analyze why fine-tuning on a single task (as shown in our experiments) failed to yield generalizable improvements, suggesting that effective data engineering would likely require a diverse range of tasks where copying shortcuts are present. Second, we will explore and at least discuss how such tasks could be systematically designed for alignment training to discourage copying behavior while maintaining model capabilities. This analysis will help contextualize our pruning method within the broader landscape of potential solutions to the copying bias problem.

---

> > > > ### Author Response · Authors · 2024-11-26
> > > >
> > > > Dear Reviewer 9Q7b!
> > > >
> > > > Thank you so much for your detailed review and the time you have already invested in discussing our work.
> > > > In our previous response, we made significant improvements to the reliability of the experiments on AlpacaEval2 (see head-to-head comparisons of pruned and unpruned models) and  introduce RULER to test retrieval abilities (as a replacement of NIAH whose reliability you questioned). I think now we have rather strong evidence that the pruning method mitigates the copying bias without compromising other capabilities. We’d love to hear your thoughts on whether these updates address your critiques.

---

> > > > > ### Comment · Reviewer_9Q7b · 2024-11-28
> > > > >
> > > > > Thanks for the author's reply. The experiments are solid. I will change my score to 6 for supporting the acceptance. Good luck!

---

> ### Comment · Reviewer_9Q7b · 2024-11-21
>
> **Experiment on Alpaca Eval2**: This is the experiment that confuses me the most. I am curious about how the authors managed to achieve such a significant improvement in the model’s performance on this dataset through neural pruning. My initial motivation for testing this issue was that if neural pruning could maintain the model’s capabilities without loss (or with minimal loss, which could be mitigated through fine-tuning), it would prove the reasonableness of this method. However, what is most intriguing is that this method not only maintained but also enhanced the model’s capabilities, achieving an improvement close to a percentage point.
>
> I believe Alpaca Eval2 does not contain many experiments that are particularly sensitive to copy bias, so such a significant improvement is quite puzzling. I suggest the authors provide more details about this experiment, such as whether the sampling parameters of the model were aligned and the number of inference runs. Additionally, specifics on the problems and answers where the pruned model outperformed the unpruned model should be included. I strongly suspect this improvement might be due to certain experimental variables, such as randomness caused by an insufficient number of runs.
>
> Also, is the "LLaMA-2" model a base one? I think the authors should use LLaMA-2 instruct version.

---

### Meta-Review · Area_Chair_PVFS · 2024-12-21

**Metareview:**

The paper proposes pruning neurons responsible for copying bias in few-shot in-context learning, identified via Integrated Gradients. While the authors report some improvements on synthetic tasks, the overall scope and novelty are limited. Key details are missing, and the paper does not situate its contribution well within the broader literature on in-context learning, including theoretical and empirical advances such as Pan et al. Comparisons to alternative interventions (e.g., better prompts, data engineering) are lacking, and real-world applicability is unclear.

Key limitations
- Evaluation focuses on synthetic or simplistic tasks. Real-world relevance is low.
- The approach draws on existing attribution-based pruning methods, so its novelty is modest.
- Missing details and the absence of code make reproducibility difficult.
- The paper neglects important prior work on ICL, which provides different ways to explain low-accuracy scenarios.
- The effect of pruning on tasks that require copying is not fully addressed.

Suggestions
- Improve literature coverage, referencing theoretical and empirical studies on ICL failures and task recognition and task learning.
- Evaluate on more diverse and many-shot tasks.
- Provide thorough methodology, release code, and compare against other potential solutions.

Overlall, the current version needs broader validation, stronger comparisons, and clearer ties to prior work before acceptance.

**Additional Comments On Reviewer Discussion:**

The authors addressed many of the reviewers' questions, but the key limitations remain the same.

---

### Decision · Program_Chairs · 2025-01-22

Reject